# PROVABLY EFFICIENT CVAR RL IN LOW-RANK MDPS

**Yulai Zhao**[*]
Princeton University
`yulaiz@princeton.edu`

**Wenhao Zhan**[*]
Princeton University
`wenhao.zhan@princeton.edu`

**Xiaoyan Hu**[*]
The Chinese University of Hong Kong
`xyhu21@cse.cuhk.edu.hk`

**Ho-fung Leung**
Independent Researcher
`ho-fung.leung@outlook.com`

**Farzan Farnia**
The Chinese University of Hong Kong
`farnia@cse.cuhk.edu.hk`

**Wen Sun**
Cornell University
`ws455@cornell.edu`

**Jason D. Lee**
Princeton University
`jasonlee@princeton.edu`

## ABSTRACT

We study risk-sensitive Reinforcement Learning (RL), where we aim to maximize the Conditional Value at Risk (CVaR) with a fixed risk tolerance $\tau$. Prior theoretical work studying risk-sensitive RL focuses on the tabular Markov Decision Processes (MDPs) setting. To extend CVaR RL to settings where state space is large, function approximation must be deployed. We study CVaR RL in low-rank MDPs with nonlinear function approximation. Low-rank MDPs assume the underlying transition kernel admits a low-rank decomposition, but unlike prior linear models, low-rank MDPs do not assume the feature or state-action representation is known. We propose a novel Upper Confidence Bound (UCB) bonus-driven algorithm to carefully balance the interplay between exploration, exploitation, and representation learning in CVaR RL. We prove that our algorithm achieves a sample complexity of $\tilde{O}\left(\frac{H^7 A^2 d^4}{\tau^2 \epsilon^2}\right)$ to yield an $\epsilon$-optimal CVaR, where $H$ is the length of each episode, $A$ is the capacity of action space, and $d$ is the dimension of representations. Computational-wise, we design a novel discretized Least-Squares Value Iteration (LSVI) algorithm for the CVaR objective as the planning oracle and show that we can find the near-optimal policy in a polynomial running time with a Maximum Likelihood Estimation oracle. To our knowledge, this is the first provably efficient CVaR RL algorithm in low-rank MDPs.

## 1 INTRODUCTION

As a widely adopted framework to deal with challenging sequential decision-making problems, Reinforcement learning (RL) (Sutton & Barto, 2018) has demonstrated many empirical successes, e.g., popular strategy games (Silver et al., 2016; 2017; Vinyals et al., 2019). However, the classical RL framework often prioritizes the maximization of the expected cumulative rewards solely, which renders it not suitable for many real-world applications that face possible failure or safety concerns. In high-stake scenarios such as autonomous driving (Isele et al., 2018; Wen et al., 2020), finance (Davis & Lleo, 2008; Wang et al., 2021; Filippi et al., 2020) and healthcare (Ernst et al., 2006; Coronato et al., 2020), optimizing only the expected cumulative rewards can produce policies that underestimate the risks associated with rare but catastrophic events. To mitigate this limitation, risk-sensitive decision-making has recently grown more popular (Hu & Leung, 2023).

---

[*]Equal contribution. Listing order is random.

To this end, risk measures like Conditional Value-at-Risk (CVaR) (Rockafellar et al., 2000) have been integrated into the RL-based systems as a performance criterion, yielding a more balanced approach that encourages policies to avoid high-risk outcomes. As a popular tool for managing risk, the CVaR metric is widely used in robust RL (Hiraoka et al., 2019), distributional RL (Achab et al., 2022), and portfolio optimization (Krokhmal et al., 2002). CVaR quantifies the expected return in the worst-case scenarios.[1] For a random variable $X$ with distribution $P$ and a confidence level $\tau \in (0, 1)$, the CVaR at confidence level $\tau$ is defined as

$$\mathsf{CVaR}_\tau(X) = \sup_{b \in \mathbb{R}} \left[ b - \cdot \mathbb{E}_{X \sim P}(b - X)^+ \right]$$

where $x^+ = \max(x, 0)$. In this paper, we consider the random variable $X$ as the cumulative reward of a specific policy where randomness comes from the MDP transitions and the policy itself. We aim to maximize this objective to capture the average worst values of the returns distribution at a certain risk tolerance level $\tau$.

The most related works to this paper are Bastani et al. (2022) and Wang et al. (2023). Specifically, Bastani et al. (2022) proved the first regret bounds for risk-sensitive RL with CVaR metric (CVaR RL), and Wang et al. (2023) improved the results to achieve the minimax optimal regret. However, their algorithms are restricted to the tabular setting, so they are inefficient when the state space is large. To overcome such limitation, we introduce function approximation to the MDP structure and consider **low-rank MDPs** (Jiang et al., 2017) in this paper.

In low-rank MDPs, the key idea is to exploit the (high-dimensional) structure in the model transitions of the MDP by representing them with a lower-dimensional structure. This is often implemented by assuming the MDP transition kernels admit a low-rank factorization, i.e., there exist two *unknown* mappings $\psi(s'), \phi(s, a)$, such that the probability of transiting to the next state $s'$ under the current state and action $(s, a)$ is $P(s'|s, a) = \psi(s')^\top \phi(s, a)$. Low-rank MDPs generalize popular RL models including tabular MDPs, linear MDPs (Jin et al., 2020) and block MDPs (Du et al., 2019; Efroni et al., 2021), and are widely applied in practice (Zhang et al., 2020; 2021; Sodhani et al., 2021; 2022).

Unlike linear MDPs, since the underlying $\psi$ and $\phi$ are unknown, we need to carefully balance representation learning, exploration, and worst-case failure stakes in low-rank MDPs. The application of non-linear function approximation further complicates the problem, for controlling model errors state and action-wise becomes much harder (Wang et al., 2023). Moreover, even within a given MDP, planning in CVaR RL is not a straightforward task as the CVaR metric exhibits a non-linear structure[2] (in stark contrast to standard risk-neutral RL), which results in extra computational overhead. All of these challenges call for a more comprehensive algorithm design for the risk-sensitive RL in low-rank MDPs.

In this paper, we aim to fill a gap in the existing body of knowledge by exploring the interplay between risk-averse RL and low-rank MDPs. We summarize our contributions as follows.

**Contributions.** First, we design an oracle-efficient representation learning algorithm called *ELA* (*REprensentation Learning for CVAR*), which optimizes the CVaR metric in low-rank MDPs. *ELA* leverages an MLE oracle to learn the model dynamics while simultaneously constructing Upper Confidence Bound (UCB)-type bonuses to encourage exploration of the unknown environment. We provide a comprehensive theoretical analysis of the algorithm, demonstrating that *ELA* would provide an $\epsilon$-optimal CVaR with $\tilde{O}(1/\epsilon^2)$ samples. To the best of our knowledge, *ELA* is the first provably sample-efficient algorithm for CVaR RL in low-rank MDPs.

Second, to improve the computational complexity of *ELA* planning, we introduce a computationally efficient planning oracle, which, when combined with *ELA*, leads to the *ELLA* (*REprensentation Learning with LSVI for CVAR*) algorithm. This algorithm leverages least-squares value iteration with discretized rewards to find near-optimal policies via optimistic simulations within the learned model. Importantly, the computational cost solely depends on the dimension of representations rather than the state space size. We show that *ELLA* requires a polynomial running time in addition to polynomial calls to the MLE oracle.

---

[1] Standard CVaR definition considers average worst-case *loss*, thus a lower value is more desirable. However, in this work, we aim to obtain a higher CVaR in the RL context as we are maximizing rewards.

[2] In RL, planning refers to determining the optimal policy and optimal value function within a given MDP.

## 2 RELATED WORK

**Low-rank MDPs.** Theoretical benefits of low-rank structure in MDPs have been broadly explored in various works (Jiang et al., 2017; Sun et al., 2019; Du et al., 2021; Sekhari et al., 2021; Huang et al., 2023). In a contextual decision process (a generic RL model with rich observations and function approximation) with a low Bellman rank, OLIVE (Jiang et al., 2017) yielded a near-optimal policy with polynomial samples. Additionally, Sun et al. (2019) introduced provably efficient algorithms based on a structural parameter called the witness rank, demonstrating that the witness rank is never larger than the Bellman rank (Jiang et al., 2017). Despite their provable efficiency, these algorithms lack computational efficiency.

Leveraging Maximum Likelihood Estimation (MLE) as its computation oracle, Flambe (Agarwal et al., 2020) proposed the first computationally efficient algorithm that was also provably efficient in low-rank MDPs. Following a similar setup, Rep-UCB (Uehara et al., 2022) improves the sample complexity dependencies of Flambe. The key to the improvements is a careful tradeoff between exploration and exploitation by combining the reward signal and exploration bonus.

**CVaR RL.** There is a long line of works studying (static) CVaR in RL, which refers to the CVaR of *accumulative reward* beyond a certain risk threshold (Chow & Ghavamzadeh, 2014; Chow et al., 2015; Tamar et al., 2015; Bastani et al., 2022; Wang et al., 2023). For tabular MDPs, much work has been done on value-based algorithms (Chow et al., 2015; Stanko & Macek, 2019). However, these results often require the planner to know the model transitions of the MDPs, which is generally infeasible. Recently, there has been a growing interest in the more general setting where the unknown model transitions are learned through online interactions (Yu et al., 2018; Bastani et al., 2022; Wang et al., 2023). However, their results are restricted to the tabular setting and cannot be combined with function approximation, where the state space is often enormous.

Prior works have also studied risk-sensitive RL in the context of other risk measures. Specifically, Du et al. (2022) proposed Iterated CVaR RL, also known as dynamic CVaR, and Chen et al. (2023) demonstrates regret guarantees for Iterated CVaR RL with general function approximations. However, iterated CVaR is intrinsically different from the static CVaR in our setting. Iterated CVaR quantifies the worst $\tau$-percent performance **at each step** of decision-making. Such a definition allows the agent to control the risk throughout the decision process tightly, whereas our setting aims to maximize the CVaR of the **total** reward. Therefore, their algorithm designs and analysis techniques do not apply to our tasks.

Our work also focuses on the static CVaR in RL. Specifically, we present the first sample-efficient algorithm for optimizing the static CVaR metric that carefully balances the interplay between risk-averse RL and low-rank MDP structure. Furthermore, we design a computationally efficient planning oracle that makes our algorithm only require polynomial running time with an MLE oracle.

## 3 PRELIMINARIES

**Notations.** We will frequently use $[H]$ to denote the set $\{1, \cdots, H\}$. Denote $\Delta(\mathcal{S})$ as the distribution over space $\mathcal{S}$. Let $U(\mathcal{A})$ be the uniform distribution over the action space. In this work, we use the standard $O(\cdot)$, $\Omega(\cdot)$ and $\Theta(\cdot)$ notations to hide universal constant factors and use $\tilde{O}(\cdot)$ to hide logarithmic factors. Please see Table 1 for a comprehensive list of notations used in this work.

### 3.1 LOW-RANK EPISODIC MDP

We consider an episodic MDP $\mathcal{M}$ with episode length $H \in \mathcal{N}$, state space $\mathcal{S}$, and a finite action space $\mathcal{A}$. At each episode $k \in [K]$, a trajectory $\tau = (s_1, a_1, s_2, a_2, \cdots, s_H, a_H)$ is generated by an agent, where (a) $s_1 \in \mathcal{S}$ is a *fixed* starting state,[3] (b) at step $h$, the agent chooses action according to a history-dependent policy $a_h \sim \pi(\cdot|\tau_{h-1}, s_h)$ where $\tau_{h-1} := (s_1, a_1, \cdots, a_{h-1})$ denotes the history and (c) the model transits to the next state $s_{h+1} \sim P_h^*(\cdot|s_h, a_h)$. The agent repeats these steps till the end of the episode. For each time step, operators $P_h^* : \mathcal{S} \times \mathcal{A} \to \Delta(\mathcal{S})$ denote the (non-stationary) transition dynamics, whereas $r_h : \mathcal{S} \times \mathcal{A} \to \Delta([0, 1])$ is the (immediate) reward distribution the

---

[3]Note that any $H$-length episodic MDP with a stochastic initial state is equivalent to an $(H + 1)$-length MDP with a dummy initial state $s_0$.

agent could receive from deploying a certain action at a specific state. We use $\Pi$ to denote the class of all history-dependent policies. Below, we proceed to the low-rank MDP definition.

**Definition 3.1** (Low-rank episodic MDP). The transition kernel $\mathcal{P}^* = \{P_h^* : \mathcal{S} \times \mathcal{A} \mapsto \Delta(\mathcal{S})\}_{h \in [H]}$ admits a low-rank decomposition with rank $d$ if there exist two embedding functions $\phi^* := \{\phi_h^* : \mathcal{S} \times \mathcal{A} \mapsto \mathbb{R}^d\}_{h \in [H]}$ and $\psi^* := \{\psi_h^* : \mathcal{S} \mapsto \mathbb{R}^d\}_{h \in [H]}$ such that

$$P_h^*(s'|s,a) = \langle \psi_h^*(s'), \phi_h^*(s,a) \rangle \tag{1}$$

where $\|\phi_h^*(s,a)\|_2 \leq 1$ for all $(h,s,a) \in [H] \times \mathcal{S} \times \mathcal{A}$, and for any function $g : \mathcal{S} \mapsto [0,1]$ and $h \in [H]$, it holds that $\|\int_{s \in \mathcal{S}} \psi_h^*(s) g(s) ds\|_2 \leq \sqrt{d}$. Low-rank episodic MDP admits such a decomposition of $\mathcal{P}^*$.

We study the function approximation setting where the state spaces $\mathcal{S}$ can be enormous and even infinite. To generalize across states, assume access to two embedding classes $\Psi \subset \{\mathcal{S} \times \mathcal{A} \to \mathbb{R}^d\}$ and $\Phi \subset \{\mathcal{S} \to \mathbb{R}^d\}$, which are used to identify the true embeddings $(\psi^*, \phi^*)$. Formally, we need the following realizability assumption, which is proven essential for obtaining performance guarantees independent of the state space size in low-rank MDPs (Agarwal et al., 2020).

**Assumption 3.2.** There exists a model class $\mathcal{F} = (\Psi, \Phi)$ such that $\psi_h^* \in \Psi, \phi_h^* \in \Phi, \forall h \in [H]$.

We expect to obtain sample complexity that scales logarithmically with the (finite) cardinality of the model class $\mathcal{F}$. Extensions to the infinite function classes with complexity measures are possible. See more discussions in (Agarwal et al., 2020).

## 3.2 RISK-SENSITIVE RL AND AUGMENTED MDP

We study risk-sensitive RL with the CVaR metric. Throughout the paper, let $\tau \in (0,1]$ be a fixed risk tolerance. First, we recall the classical definition: for a random variable $X \in [0,1]$ is from distribution $P$, the *conditional-value-at-risk* (CVaR) corresponding to the risk tolerance $\tau$ is defined as

$$\mathsf{CVaR}_\tau(X) := \sup_{c \in [0,1]} \left\{ c - \tau^{-1} \cdot \mathbb{E}_{X \sim P}[(c - X)^+] \right\} \tag{2}$$

where $(x)^+ := \max(x, 0)$ for any $x \in \mathbb{R}$. Interestingly, the supremum in the expression is achieved when $c$ is set as the $\tau$-th percentile (unknown before planning), also known as *value-at-risk* (VaR), i.e., $x_\tau = \inf\{x \in \mathbb{R} : P(X \leq x) \geq \tau\}$.

In risk-sensitive RL, $X$ represents the stochastic cumulative reward accumulated over $H$ successive actions and state transitions: $X = \sum_{h=1}^H r_h$. This multi-step nature of risk-sensitive RL brings the dynamic planning structure to the setting and makes it difficult. We may make an intuitive insight of $c$ by considering it as a *initial budget*, which affects the agent's action selection and needs to be carefully managed during the planning. Particularly, after receiving a random reward $r_h$ at each timestep $h \in [H]$, the learner deducts it from the current budget, i.e., $c_{h+1} = c_h - r_h$ where $c_h$ is the remaining budget and $c_1 = c$. The main task of risk-sensitive RL is to interplay carefully between policy planning, exploration, and budget control. Inspired by this observation, we incorporate the available budget as an additional state variable that could impact the agent's choice of action. Formal descriptions of the augmented MDP are introduced below.

**Augmented MDP.** We introduce the augmented MDP framework (Bäuerle & Ott, 2011) to study CVaR RL, which augments the state space $\mathcal{S}$ in classic episodic MDP to $\mathcal{S}_{\text{Aug}} = \mathcal{S} \times [0, H]$ that includes the budget variable as an additional state. The transition kernel of the state and the immediate reward distribution are the same as the original MDP $\mathcal{M}$. Inspired by the observation above, we consider policies defined on the augmented state space $\Pi_{\text{Aug}} := \{\pi : \mathcal{S}_{\text{Aug}} \to \Delta(\mathcal{A})\}$.

In the augmented MDP, the agent rolls out an augmented policy $\pi \in \Pi_{\text{Aug}}$ with initial budget $c_1 \in [0, H]$ as follows. At the beginning of an episode, the agent observes the augmented state $(s_1, c_1)$, selects action $a_1 \sim \pi(\cdot|s_1, c_1)$, receives reward $r_1 \sim r_1(s_1, a_1)$, and transits to $s_2 \sim P_1^*(\cdot|s_1, a_1)$. Most importantly, the available budget is also updated: $c_2 = c_1 - r_1$. The agent then chooses an action based on the $(s_2, c_2)$ pair. The procedure repeats for $H$ times until the end of the episode. For any $\pi \in \Pi_{\text{Aug}}$, the (augmented) $Q$-function is defined as

$$Q_{h,\mathcal{P}^*}^\pi(s,c,a) := \mathbb{E}_{\pi,\mathcal{P}^*}\left[\left(\left(c_h - \sum_{t=h}^H r_t(s_t, a_t)\right)^+ \Big| s_h = s, c_h = c, a_h = a\right)\right].$$

for any $(h, s, c, a) \in [H] \times \mathcal{S} \times [0, H] \times \mathcal{A}$ and the (augmented) value function is defined as

$$V_{h,\mathcal{P}^*}^\pi(s, c) := \mathop{\mathbb{E}}_{a \sim \pi_h(\cdot|s,c)} \left[ Q_{h,\mathcal{P}^*}^\pi(s, c, a) \right] \tag{3}$$

for any $(h, s, c) \in [H] \times \mathcal{S} \times [0, H]$. The Bellman equation is given by

$$Q_{h,\mathcal{P}^*}^\pi(s, c, a) = \mathop{\mathbb{E}}_{s' \sim P_h^*(\cdot|s,a), r \sim r_h(s,a)} V_{h+1}^\pi(s', c - r).$$

**Goal metric.** In this paper, we aim to find the optimal history-dependent policy to maximize the CVaR objective, i.e.,

$$\mathsf{CVaR}_\tau^* := \max_{\pi \in \Pi} \mathsf{CVaR}_\tau(R(\pi)) = \sup_{c \in [0,H]} \left\{ c - \tau^{-1} \cdot \min_{\pi \in \Pi} \mathbb{E}[(c - R(\pi))^+] \right\}$$

where $R(\pi)$ is the random cumulative reward of policy $\pi$ in $\mathcal{M}$. Nevertheless, it is known that $\min_{\pi \in \Pi}\{\tau^{-1} \cdot \mathbb{E}[(c - R(\pi))^+]\}$ can be attained by an augmented policy $\pi^* \in \Pi_{\mathrm{Aug}}$ with initial budget $c$ (Wang et al., 2023). Thus we can indeed focus on searching within $\pi \in \Pi_{\mathrm{Aug}}$:

$$\mathsf{CVaR}_\tau^* = \max_{c \in [0,H]} \left\{ c - \tau^{-1} \cdot \min_{\pi \in \Pi_{\mathrm{Aug}}} V_{1,\mathcal{P}^*}^\pi(s_1, c) \right\} := \mathsf{CVaR}_\tau(R(\pi^*, c^*)) \tag{4}$$

where $\pi^* \in \Pi_{Aug}$ is the optimal augmented policy and $c^* \in [0, H]$ is the optimal initial budget, and we overload $R(\pi, c)$ to denote the stochastic cumulative reward when the agent rolls out the augmented policy $\pi$ with initial budget $c$ in augmented MDP.

## 4 ALGORITHM

In this section, we present the *ELA* algorithm for risk-sensitive RL in low-rank MDPs. The pseudocode is listed in Algorithm 1. The algorithm is iterative in nature, where the $k$-th episode proceeds in three folds: (1) we collect new data by rolling in with the exploration policy $\pi^{k-1}$ starting from the initial budget $c^{k-1}$. (2) Then, all transitions collected so far are used in two aspects. First, we pass all transition tuples to the MLE oracle (Line 11). The MLE oracle returns embedding functions $(\widehat{\psi}_h, \widehat{\phi}_h)$ for each $h$, which determine the model. Second, we compute the exploration bonus using the latest learned representation $\widehat{\phi}$. (3) The algorithm performs VI on the learned model with the bonus-enhanced reward signal to obtain the exploration policy-budget pair $(\pi^k, c^k)$ we use in the next iteration. After $K$ iterations, we output the current model and all policy-budget pairs. Next, we illustrate more details about these steps.

### 4.1 DATA COLLECTION

During training, we collect two (disjoint) sets of transition tuples to compute the bonus terms and estimate the transition kernels. These two datasets are different in their (marginalized) distributions and facilitate the regret analysis in Section 4.3. Next, we clarify how data are collected and added to the two sets.

To collect a new transition tuple for each dataset $\mathcal{D}_h$ and $\widetilde{\mathcal{D}}_h$ ($\forall h \in [H - 1]$), at the $k$-th iteration, the algorithm roll outs policy $\pi^{k-1}$ with initial budget $c^{k-1}$ to obtain two trajectories. The difference occurs at the $(h-1)$-th timestep. Particularly, to obtain $(s_h, a_h, \tilde{s}_{h+1})$ for $\mathcal{D}_h$, the algorithm *keeps on* to execute policy $\pi^{k-1}$ for one more step and observes state $s_h$. Then, a uniform action $a_h \sim U(\mathcal{A})$ is taken to receive the next-state $\tilde{s}_{h+1} \sim P_h^*(\cdot|s_h, a_h)$. However, to obtain $(\tilde{s}_h, \tilde{a}_h, s'_{h+1})$ for $\widetilde{\mathcal{D}}_h$, the algorithm takes *two consecutive* uniform actions at timesteps $h - 1$ and $h$, i.e., $a_{h-1} \sim U(\mathcal{A}), \tilde{s}_h \sim P_{h-1}^*(\cdot|s_{h-1}, a_{h-1}), \tilde{a}_h \sim U(\mathcal{A})$, and receives state $s'_{h+1}$. Intuitively, dataset $\mathcal{D}_h$ reveals the behavior of the exploration policy-budget pair $(\pi^k, c^k)$ up to the $h$-th timestep, which facilitates the design of bonus terms (cf. Lemma C.2). Meanwhile, dataset $\widetilde{\mathcal{D}}_h$ enhances the exploration and leads to improved estimates of transition kernels (cf. Lemma E.3).

At first sight, the data collection procedure requires $2(H - 1)$ trajectories per iteration (i.e., one trajectory for $\mathcal{D}_h$ and one trajectory for $\widetilde{\mathcal{D}}_h$). To save sample (trajectory) complexity, we collect

transition tuples for $\mathcal{D}_h$ and $\widetilde{\mathcal{D}}_h$ at the $h, (h-1)$ steps in one go, respectively (Lines 8-10). Therefore, the algorithm only requires $H$ trajectories per iteration. For both sets, new transition tuples are concatenated with the existing data to perform representation learning, i.e., learning a factorization and a representation by MLE (Line 11).

## 4.2 REPRESENTATION LEARNING AND BONUS-DRIVEN VALUE ITERATION

**MLE oracle.** In the function approximation setting, the agent must estimate the model structure as accurately as possible. Collected transition tuples are used to compute model transitions through the MLE oracle. As a general approach used to estimate the parameters of a probability model, MLE is also gaining more focus in low-rank MDPs (Agarwal et al., 2020; Uehara et al., 2022).

**Value Iteration.** Based on the learned model, the algorithm runs Value-Iteration (VI) with the exploration bonus term. In risk-sensitive RL, for any $\pi \in \Pi_{Aug}$ and $(h, s, c) \in [H] \times \mathcal{S} \times [0, H]$, we define value function enhanced with exploration bonus $b_h : \mathcal{S} \times \mathcal{A} \to \mathbb{R}$ as

$$V_{h,\mathcal{P},b}^{\pi}(s,c) := \mathbb{E}_{\pi,\mathcal{P}} \left[ \left( c_h - \sum_{h'=h}^{H} r_{h'}(s_{h'}, a_{h'}) \right)^+ - \sum_{h'=h}^{H} b_{h'}(s_{h'}, a_{h'}) \middle| s_h = s, c_h = c \right] \quad (5)$$

where we deduct the exploration bonus term because the agent desires to *minimize* the value function (4) in risk-sensitive RL. Such a value function is used to perform VI and update policy on the learned model $\widehat{P}$, since the learner has no prior knowledge of the *real* model transitions. Therefore, obtaining an accurate estimation of the model determines the quality of the output policy.

In Algorithm 1, we assume the learner has access to exact VI (Line 15). However, we remark that this step is not computationally efficient due to the continuity of $c$ and potentially large state space $\mathcal{S}$. To overcome such a computational barrier arising from the nature of controlling risk while planning, in Section 5, we provide a computationally efficient planning oracle that performs LSVI with discretized reward function (with sufficiently high precision). We rigorously prove that we only need polynomial running time with the MLE oracle to output a near-optimal CVaR.

## 4.3 MAIN RESULTS

In this subsection, we are ready to demonstrate our theoretical guarantees that characterize the regret and sample complexity bounds.

**Theorem 4.1.** *Fix $\delta \in (0, 1)$. Set the parameters in Algorithm 1 as:*

$$\alpha^k = O\left( \sqrt{H^2(|\mathcal{A}| + d^2)} \log\left( \frac{|\mathcal{F}| H k}{\delta} \right) \right), \quad \lambda^k = O\left( d \log\left( \frac{|\mathcal{F}| H k}{\delta} \right) \right).$$

*We have two equivalent interpretations of the theoretical results. In terms of PAC bound, with probability at least $1 - \delta$, the regret is bounded by*

$$\sum_{k=1}^{K} \mathsf{CVaR}_\tau^* - \mathsf{CVaR}_\tau(R(\pi^k, c^k)) = \tilde{O}\left( \tau^{-1} H^3 A d^2 \sqrt{K} \cdot \sqrt{\log(|\mathcal{F}|/\delta)} \right).$$

*Alternatively, we can interpret in terms of sample complexity: w.p. at least $1 - \delta$, to present an $\epsilon$-optimal policy and budget pair s.t. $\mathsf{CVaR}_\tau^* - \mathsf{CVaR}_\tau(R(\widehat{\pi}, \widehat{c})) \le \epsilon$. The total number of trajectories required is upper bounded by*

$$\tilde{O}\left( \frac{H^7 A^2 d^4 \log(|\mathcal{F}|/\delta)}{\tau^2 \epsilon^2} \right).$$

*Proof.* The proof can be found in Appendix C. □

In Theorem 4.1, we present the first regret/sample complexity bounds for CVaR RL with function approximation, in which exploring the unknown action/space spaces posits extra difficulty. We explicitly characterize the number of samples required to output an $\epsilon$-optimal CVaR in terms of the length of the episode $H$, the action space size $A$, the representation dimension $d$, and confidence level $\tau$. The theorem has no explicit dependence on state space $\mathcal{S}$, proving nice guarantees even in

---

**Algorithm 1** *ELA*

---

**Require:** Risk tolerance $\tau \in (0, 1]$, number of iterations $K$, parameters $\{\lambda^k\}_{k \in [K]}$ and $\{\alpha^k\}_{k \in [K]}$, models $\mathcal{F} = \{\Psi, \Phi\}$, failure probability $\delta \in (0, 1)$.

1: Set datasets $\mathcal{D}_h, \widetilde{\mathcal{D}}_h \leftarrow \emptyset$ for each $h \in [H-1]$.
2: Initialize the exploration policy $\pi^0 \leftarrow \{\pi_h^0(s, c) = U(\mathcal{A}), \text{for any } (s, c) \in \mathcal{S} \times [0, H]\}_{h \in [H]}$.
3: Initialize the budget $c^0 \leftarrow 1$.
4: **for** iteration $k = 1, \ldots, K$ **do**
5:     Collect a tuple $(\tilde{s}_1, \tilde{a}_1, s_2')$ by taking $\tilde{a}_1 \sim U(\mathcal{A})$, $s_2' \sim P_1^*(\cdot | \tilde{s}_1, \tilde{a}_1)$.
6:     Update $\widetilde{\mathcal{D}}_1 \leftarrow \widetilde{\mathcal{D}}_1 \cup \{(\tilde{s}_1, \tilde{a}_1, s_2')\}$.
7:     **for** $h = 1, \cdots, H-1$ **do**
8:         Collect two transition tuples $(s_h, a_h, \tilde{s}_{h+1})$ and $(\tilde{s}_{h+1}, \tilde{a}_{h+1}, s_{h+2}')$ by first rolling out $\pi^{k-1}$ starting from $(s_1, c^{k-1})$ into state $s_h$, taking $a_h \sim U(\mathcal{A})$, and receiving $\tilde{s}_{h+1} \sim P_h^*(\cdot | s_h, a_h)$, then taking $\tilde{a}_{h+1} \sim U(\mathcal{A})$ and receiving $s_{h+2}' \sim P_{h+1}^*(\cdot | \tilde{s}_{h+1}, \tilde{a}_{h+1})$.
9:         Update $\mathcal{D}_h \leftarrow \mathcal{D}_h \cup \{(s_h, a_h, \tilde{s}_{h+1})\}$.
10:       Update $\widetilde{\mathcal{D}}_{h+1} \leftarrow \widetilde{\mathcal{D}}_{h+1} \cup \{(\tilde{s}_{h+1}, \tilde{a}_{h+1}, s_{h+2}')\}$ if $h \leq H-2$.
11:       Learn representations via MLE

$$\widehat{P}_h := (\widehat{\psi}_h, \widehat{\phi}_h) \leftarrow \arg \max_{(\psi, \phi) \in \mathcal{F}} \sum_{(s_h, a_h, s_{h+1}) \in \{\mathcal{D}_h + \widetilde{\mathcal{D}}_h\}} \log \langle \psi(s_{h+1}), \phi(s_h, a_h) \rangle$$

12:       Update empirical covariance matrix $\widehat{\Sigma}_h = \sum_{(s,a) \in \mathcal{D}_h} \widehat{\phi}_h(s, a) \widehat{\phi}_h(s, a)^\top + \lambda^k I_d$.
13:       Set the exploration bonus:

$$\widehat{b}_h(s, a) \leftarrow \begin{cases} \min\left(\alpha^k \sqrt{\widehat{\phi}_h(s, a) \widehat{\Sigma}_h^{-1} \widehat{\phi}_h(s, a)^\top}, 2\right) & h \leq H-2 \\ 0 & h = H-1 \end{cases}$$

14:     **end for**
15:     Run Value-Iteration (VI) and obtain $c^k \leftarrow \arg \max_{c \in [0,H]} \left\{ c - \tau^{-1} \min_\pi V_{1, \widehat{\mathcal{P}}, \widehat{b}}^\pi(s_1, c) \right\}$.
16:     Set $\pi^k \leftarrow \arg \min_\pi V_{1, \widehat{\mathcal{P}}, \widehat{b}}^\pi(s_1, c^k)$.
17: **end for**
**Ensure:** Uniformly sample $k$ from $[K]$, return $(\widehat{\pi}, \widehat{c}) = (\pi^k, c^k)$.

---

the infinite-state setting. As far as we are concerned, the only existing theoretical guarantees in the field of risk-averse CVaR RL were provided in the tabular setting (Bastani et al., 2022; Wang et al., 2023), where representations are known. Moreover, these results explicitly depend on $|\mathcal{S}|$, thus can not apply to the function approximation setting with enormous state space. Given the above, our work accomplishes a great leap by incorporating exploration in the comprehensive function approximation setting, which evidently better aligns with real-world applications than the tabular and/or linear MDP settings.

As for the sample complexity, the dependencies on $A$ and $d$ match the same rates as the analysis of risk-neutral RL in low-rank MDP (Uehara et al., 2022), showing our algorithm overcomes the extra hardness of balancing exploration and budget control even with a large action space. Our sample complexity matches the dependency on $\tau$ with the rates in the CVaR-UCBVI algorithm with the Hoeffding bonus (Wang et al., 2023) and UCB algorithm (Bastani et al., 2022). On the other hand, the sample complexity enjoys an $\Omega\left(\frac{1}{\tau \epsilon^2}\right)$ lower bound(Wang et al., 2023, Theorem 3.1). Tightening the dependency on $\tau$ is left as future work.

Our theoretical guarantees have a slightly worse dependency on $H^7$ while Rep-UCB (Uehara et al., 2022) scales as $H^5$, since we do not assume the trajectory reward is normalized to 1. If we adopt this reward normalization, our PAC bound can be improved to $\tilde{O}(H^5)$, and thus matching the rate. In addition, we point out that it might be possible to reduce another $H^2$ dependency by utilizing a Bernstein-type bonus in the algorithm instead of the Hoeffding-type bonus, which is left for future work.

## 5 PLANNING ORACLE: *CVaR-LSVI*

In Algorithm 1, we need to calculate $c^k \leftarrow \arg\max_{c\in[0,H]}\left\{c - \tau^{-1}\min_\pi V_{1,\widehat{\mathcal{P}},\widehat{b}}^\pi(s_1,c)\right\}$ (Line 15), which is not naively computationally efficient since the objective $c - \tau^{-1}\min_\pi V_{1,\widehat{\mathcal{P}},\widehat{b}}^\pi(s_1,c)$ is not concave. In this section, we introduce a feasible planning oracle for this step and provide the corresponding theoretical guarantees. For simplicity, we assume the reward distribution $r$ is discrete and $r_h(s,a)$ only takes value in $iv$ where $v > 0$ and $0 \leq i \leq \lceil 1/v \rceil$. For a continuous $r$, we can discretize it with sufficiently high precision so that all the analysis still applies. The details are deferred to Appendix B.

Since the supremum of the CVaR objective (2) is attained at $\tau$-th quantile of the cumulative reward distribution, which is also discrete when the reward $r_h$ is discrete, we can only search $c^k$ within the grid in Line 15 of Algorithm 1:

$$c^k \leftarrow v \cdot \operatorname*{arg\,max}_{0\leq i\leq \lceil H/v\rceil}\left\{iv - \tau^{-1}\min_{\pi\in\Pi_{Aug}} V_{1,\widehat{\mathcal{P}},\widehat{b}}^\pi(s_1,iv)\right\}.$$

Nevertheless, if we run standard value iteration to calculate $\min_{\pi\in\overline{\Pi}_{Aug}} V_{1,\widehat{\mathcal{P}},\widehat{b}}^\pi(s_1,iv)$, our sample complexity will scale with the size of the state space $|\mathcal{S}|$, which can be infinite in low-rank MDPs. To circumvent such dependency on $|\mathcal{S}|$, we introduce a novel LSVI-UCB algorithm for the CVaR objective in this subsection, called *CVaR-LSVI*. In the following discussion, we fix $i_1$ where $0 \leq i_1 \leq \lceil H/v\rceil$ and aim at calculating $\min_{\pi\in\Pi_{Aug}} V_{1,\widehat{\mathcal{P}},\widehat{b}}^\pi(s_1,i_1v)$. We will drop the subscript $\widehat{\mathcal{P}}$ and $\widehat{b}$ when it is clear from the context. We use $V_h^*$ to denote $\min_{\pi\in\Pi_{Aug}} V_{h,\widehat{\mathcal{P}},\widehat{b}}^\pi$ and $(\widehat{\mathcal{P}},r)$ to denote the MDP model whose transition is $\widehat{\mathcal{P}}$ and reward distribution is $r$.

First, recall that the discrete reward distribution $r_h(s,a)$ only takes values of $iv$ where $0 \leq i \leq \lceil 1/v\rceil$. This means that it is always linear with respect to a $(\lceil 1/v\rceil + 1)$-dimension vector:

$$r_h(iv|s,a) = \langle \phi_{h,r}(s,a), \psi_{h,r}(iv)\rangle,$$

where $(\phi_{h,r}(s,a))_i = r_h(iv|s,a)$ and $\psi_{h,r}(iv) = e_i$ for all $0 \leq i \leq \lceil 1/v\rceil$ and $s \in \mathcal{S}, a \in \mathcal{A}, h \in [H]$. Since $\widehat{P}_h(s'|s,a) = \langle\widehat{\phi}_h(s,a), \widehat{\psi}_h(s')\rangle$, this implies that for all $s, s' \in \mathcal{S}, a \in \mathcal{A}, h \in [H], 0 \leq i \leq \lceil 1/v\rceil$, we have

$$\widehat{P}_h(s'|s,a)r_h(iv|s,a) = \langle \overline{\phi}_h(s,a), \overline{\psi}_h(s',iv)\rangle,$$

where $\overline{\phi}_h(s,a) = \widehat{\phi}_h(s,a) \otimes \phi_{h,r}(s,a)$ and $\overline{\psi}_h(s',iv) = \widehat{\psi}_h(s,a) \otimes \psi_{h,r}(iv)$.

The linearity of transition and reward implies that we can utilize LSVI to compute $Q_h^\pi(s,c,a)$. More specifically, we propose an iterative algorithm consisting of the following steps:

- **Step 1: Ridge Regression.** In the $t$-th iteration, denote the trajectories collected before $t$-th iteration by $\{(s_h^j, a_h^j, \bar{r}_h^j)_{h=1}^H\}_{j=1}^{t-1}$. Let $V_{H+1}^t(s,iv) = iv$ for all $s \in \mathcal{S}$ and $0 \leq i \leq \lceil H/v\rceil$. From $h = H$ to $h = 1$, for each $0 \leq i \leq \lceil H/v\rceil$, we first compute:

$$w_h^t(iv) \leftarrow (\Lambda_h^t)^{-1}\sum_{j=1}^{t-1}\overline{\phi}_h(s_h^j, a_h^j) \cdot \left[V_{h+1}^t(s_{h+1}^j, iv - r_h^j)\right],$$

where $\Lambda_h^t = \lambda I + \sum_{j=1}^{t-1}\overline{\phi}_h(s_h^j, a_h^j)(\overline{\phi}_h(s_h^j, a_h^j))^\top$.

Then for any $j \in [t-1], a \in \mathcal{A}$ and $0 \leq i \leq \lceil H/v\rceil$, we can estimate the value function $V_h^t(s_h^j, iv)$ as:

$$Q_h^t(s_h^j, iv, a) = \text{Clip}_{[-H,H]}\left(-\widehat{b}_h(s_h^j, a) + (\overline{\phi}_h(s_h^j, a))^\top w_h^t(iv) - \beta\|\overline{\phi}_h(s_h^j, a)\|_{(\Lambda_h^t)^{-1}}\right),$$

$$V_h^t(s_h^j, iv) = \min_{a\in\mathcal{A}} Q_h^t(s_h^j, iv, a).$$

Note that although we do not compute $Q_h^t(s, iv, a)$ for all $s \in \mathcal{S}$ (which will incur computation cost scaling with $|\mathcal{S}|$), they can be implicitly expressed via $w_h^t(iv)$, i.e., for all $s \in \mathcal{S}, a \in \mathcal{A}, 0 \leq i \leq \lceil H/v\rceil$, we know

$$Q_h^t(s, iv, a) = \text{Clip}_{[-H,H]}\left(-\widehat{b}_h(s, a) + (\overline{\phi}_h(s, a))^\top w_h^t(iv) - \beta\|\overline{\phi}_h(s, a)\|_{(\Lambda_h^t)^{-1}}\right). \quad (6)$$

- **Step 2: Sample Collection.** In the $t$-th iteration, simulate the greedy policy $\widetilde{\pi}^t$ (w.r.t. the estimated Q function $Q_h^t(s, iv, a)$) with the initial budget $c_1 = i_1 v$ in the MDP model $(\widehat{\mathcal{P}}, r)$ and collect a trajectory $(s_h^t, a_h^t, r_h^t)_{h=1}^H$. Then, go back to the first step.

- **Step 3: Policy Evaluation.** After repeating the above two steps for $T_1$ iterations, we simulate each policy $\widetilde{\pi}^t$ with initial budget $c_1 = i_1 v$ in $(\widehat{\mathcal{P}}, r)$ for $T_2$ episodes. Suppose the collected trajectories are $\left\{ \left( s_h^{t,j}, a_h^{t,j}, r_h^{t,j} \right)_{h=1}^H \right\}_{j=1}^{T_2}$ and we estimate the empirical value function of $\widetilde{\pi}^t$ as follows:

$$\widehat{V}_1^{\widetilde{\pi}^t}(s_1, i_1 v) = \frac{1}{T_2} \sum_{j=1}^{T_2} \left( i_1 v - \sum_{h=1}^H r_h^{t,j}(s_h^{t,j}, a_h^{t,j}) \right)^+ - \sum_{h=1}^H \widehat{b}_h(s_h^{t,j}, a_h^{t,j}).$$

Then we simply use $\max_{t \in [T_1]} \widehat{V}_1^{\widetilde{\pi}^t}(s_1, i_1 v)$ as a surrogate for $V_1^*(s_1, i_1 v)$.

The details of *CVaR-LSVI* are stated in Algorithm 2 (cf. Appendix B). Note that in Line 16 of Algorithm 1, we can also use the above *CVaR-LSVI* algorithm to compute $\pi^k$. Combining Algorithm 1 and 2, we can derive a computationally efficient algorithm, called *ELLA*, for CVaR objective in low-rank MDPs, which is shown in Algorithm 3 (cf. Appendix B). Now, we are ready present the computational complexity of *ELLA*. Particularly, the following theorem characterizes the computational cost for finding an $\epsilon$-optimal policy:

**Theorem 5.1** (Informal). *Let the parameters in Algorithm 1 and 2 take appropriate values, then we have with probability at least $1 - \delta$ that $\mathsf{CVaR}_\tau^* - \mathsf{CVaR}_\tau(R(\widehat{\pi}, \widehat{c})) \leq \epsilon$ where $(\widehat{\pi}, \widehat{c})$ is the returned policy and initial budget by Algorithm 3. In total, the sample complexity is upper bounded by $\tilde{O}\left( \frac{H^7 A^2 d^4 \log \frac{|\mathcal{F}|}{\delta}}{\tau^2 \epsilon^2} \right)$. The MLE oracle is called $\tilde{O}\left( \frac{H^7 A^2 d^4 \log \frac{|\mathcal{F}|}{\delta}}{\tau^2 \epsilon^2} \right)$ times and the rest of the computation cost is $\tilde{O}\left( \frac{H^{19} A^3 d^{12} \log \frac{|\mathcal{F}|}{\delta}}{v^{10} \tau^6 \epsilon^6} \right)$.*

Theorem 5.1 is a special case of the continuous reward setting, whose formal statement is in Theorem B.3 and the proof is deferred to Appendix D. Theorem 5.1 indicates that Algorithm 3 is able to find a near-optimal policy with polynomial sample complexity and polynomial computational complexity given an MLE oracle. Note that calling *CVaR-LSVI* in Line 17 and 20 of Algorithm 3 will not increase the sample complexity because we are only simulating with a known model $(\widehat{\mathcal{P}}, \overline{r})$ and do not need to interact with the ground-truth environment.

## 6 CONCLUDING REMARKS

The paper proposes *ELA*, the first provably efficient algorithm for risk-sensitive reinforcement learning with the CVaR objective in low-rank MDPs. *ELA* achieves a sample complexity of $\tilde{O}(1/\epsilon^2)$ to find an $\epsilon$-optimal policy. To improve computational efficiency, we propose the *ELLA* algorithm, which leverages least-squares value iteration upon discretized reward as the planning oracle. Given the MLE oracle, we prove this algorithm only requires a polynomial computational complexity.

Regarding future work, we believe establishing lower bounds for CVaR RL in low-rank MDPs is an exciting and challenging direction due to the complexity of the risk landscape and the non-linearity in function approximation methods used. We point out that even for standard episodic RL, which is a well-studied area, we have only recently seen some results on sample lower bounds in the context of low-rank MDPs (Cheng et al., 2023; Zhao et al., 2023). However, these results do not directly apply to the CVaR risk landscape. We suppose the sample complexity lower bound may take the form of $\Omega\left( \frac{HAd}{\tau \epsilon^2} \right)$ according to the lower bounds in low-rank MDPs (Cheng et al., 2023; Zhao et al., 2023) and tabular CVaR RL (Wang et al., 2023), which implies that our algorithm has a loose dependency on $H$ and $\tau$. Such a gap might be alleviated if we use the Hoeffding-type bonus instead of the Bernstein-type bonus in the bonus-driven exploration. We leave it as future work to tighten the dependencies.

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

# A NOTATIONS

Table 1: List of Notations

**Basics**

| | |
|---|---|
| $A = \|\mathcal{A}\|$ | cardinality of the action space |
| $[H] = \{1, \cdots, H\}$ | |
| $\Delta(\cdot)$ | probability simplex |
| $Z^{\pi}_{\mathcal{P},c_1}$ | return distribution of rolling $\pi$ with initial budget $c_1$ in an MDP with $\mathcal{P}$ |
| $V^{\pi}_{h,\mathcal{P},b} : \mathcal{S} \times [0,H] \mapsto [0,H]$ | defined in (5) |
| $d^{(\pi,c_1)}_{h,\mathcal{P}}(s), d^{(\pi,c_1)}_{h,\mathcal{P}}(s,a)$ | occupancy of $s$ and $(s,a)$ at step $h$ when rolling $\pi$ with initial budget $c_1$ in an MDP with $\mathcal{P}$ |
| $d^{(\pi,c_1)}_h(s) = d^{(\pi,c_1)}_{h,\mathcal{P}^*}(s), d^{(\pi,c_1)}_h(s,a) = d^{(\pi,c_1)}_{h,\mathcal{P}^*}(s,a)$ | occupancy of $s$ and $(s,a)$ at step $h$ when rolling $\pi$ with initial budget $c_1$ in the (true) environment |

**In-Algorithm**

Initialization

| | |
|---|---|
| $\mathcal{F} = \{\Psi, \Phi\}$ | model class |
| $\lambda^k = O(d\log(\|\mathcal{F}\|Hk/\delta))$ | regularizer at iteration $k$ |
| $\alpha^k = \sqrt{H^2(A+d^2)\log(\|\mathcal{F}\|Hk/\delta)}$ | parameter at iteration $k$ |

Datasets (at the end of the $k$-th episode)

| | |
|---|---|
| $\mathcal{D}^k_h = \{(s^i_h, a^i_h, \tilde{s}^i_{h+1})\}_{i \in [k]}$ | $s^i_h \sim d^{\pi^{i-1}}_{h,c^{i-1}}, a^i_h \sim U(\mathcal{A})$ |
| $\widetilde{\mathcal{D}}^k_h = \{(\tilde{s}^i_h, \tilde{a}^i_h, s'^i_{h+1})\}_{i \in [k]}$ | $\tilde{s}^i_h \sim d^{\pi^{i-1}}_{h-1,c^{i-1}} \times U(\mathcal{A}) \times P^*_h, \tilde{a}^i_h \sim U(\mathcal{A})$ |

Estimations (at the end of the $k$-th episode)

| | |
|---|---|
| $\{(\widehat{\psi}^k_h, \widehat{\phi}^k_h)\}_{h \in [H]}$ | learned representations |
| $\widehat{\mathcal{P}}^k = \{\widehat{P}^k_h\}_{h \in [H]}$ | empirical transition kernel |
| $\widehat{\Sigma}^k_h = \sum_{(s,a) \in \mathcal{D}^k_h} \widehat{\phi}^k_h (\widehat{\phi}^k_h)^\top + \lambda^k I$ | empirical covariance matrix |
| $\widehat{b}^k = \{\widehat{b}^k_h\}_{h \in [H]}$ | bonus term |

**In-Analysis**

| | |
|---|---|
| $\rho^k_h(s) = \frac{1}{k} \sum_{i=0}^{k-1} d^{\pi^i}_{h,c^i}(s)$ | occupancy of $s$ in dataset $\mathcal{D}^k_h$ |
| $\rho^k_h(s,a) = \frac{1}{k} \sum_{i=0}^{k-1} d^{\pi^i}_{h,c^i}(s,a)$ | |
| $\eta^k_h(s) = \sum_{s',a'} \rho^k_{h-1}(s')U(a')P^*_{h-1}(s\|s',a')$ | occupancy of $s$ in dataset $\widetilde{\mathcal{D}}^k_h$ |
| $\Sigma_{\rho^k_h \times U(\mathcal{A}),\phi} = k\mathbb{E}_{s \sim \rho^k_h, a \sim U(\mathcal{A})}[\phi\phi^\top] + \lambda^k I$ | |
| $\Sigma_{\rho^k_h,\phi} = k\mathbb{E}_{(s,a) \sim \rho^k_h}[\phi\phi^\top] + \lambda^k I$ | |
| $\widehat{\Sigma}^k_{h,\phi} = k\mathbb{E}_{(s,a) \sim \mathcal{D}^k_h}[\phi\phi^\top] + \lambda^k I$ | unbiased estimate of $\Sigma_{\rho^k_h \times U(\mathcal{A}),\phi}$ |
| $\zeta^k = \log(\|\mathcal{F}\|Hk/\delta)/k$ | |
| $f^k_h(s,a) = \|\widehat{P}^k_h(\cdot\|s,a) - P^*_h(\cdot\|s,a)\|_1$ | estimation error in $L_1$ norm |
| $\omega^{(\pi,c_1)}_{h,\mathcal{P}}(\cdot\|s,a)$ | the distribution of the remaining budget at $h$ for any $(s,a)$ when rolling out $(\pi,c_1)$ in an MDP with $\mathcal{P}$ |
| $\omega^{(\pi,c_1)}_h(\cdot\|s,a) = \omega^{(\pi,c_1)}_{h,\mathcal{P}^*}(\cdot\|s,a)$ | the distribution of the remaining budget at $h$ for any $(s,a)$ when rolling out $(\pi,c_1)$ in the true MDP |

# B  *ELLA* FOR CONTINUOUS REWARD

Now, we extend the analysis in Section 5 to continuous reward distribution.

## B.1  DISCRETIZED REWARD

Inspired by Wang et al. (2023), we discretize the reward $r_h$ and the budget $c_h$ at each step. In this way, we only need to plan over a finite grid. More specifically, suppose the precision is $\upsilon > 0$, then we round up the reward $r$ to $U(r) := \lceil r/\upsilon \rceil \upsilon$. In the following discussion, we use $\overline{\mathcal{M}}$ to denote the MDP which shares the same transition as $\mathcal{M}$ while its reward $\overline{r}$ is discretized from the original reward distribution $r$ of $\mathcal{M}$, i.e., $\overline{r} = U(r)$. Since the supremum of the CVaR objective (2) is attained at $\tau$-th quantile of the return distribution, which is also discretized in $(\widehat{\mathcal{P}}, \overline{r})$, we can only search $c^k$ within the grid for $(\widehat{\mathcal{P}}, \overline{r})$ in Line 15 of Algorithm 1:

$$c^k \leftarrow \upsilon \cdot \underset{0 \le i \le \lceil H/\upsilon \rceil}{\arg\max} \left\{ i\upsilon - \tau^{-1} \min_{\pi \in \overline{\Pi}_{Aug}} \overline{V}^{\pi}_{1, \widehat{\mathcal{P}}, \widehat{b}}(s_1, i\upsilon) \right\}, \tag{7}$$

where $\overline{\Pi}_{Aug}$ is the augmented policy class of augmented $\overline{\mathcal{M}}$ and $\overline{V}^{\pi}_{h, \mathcal{P}, b}$ is the value function of $(\mathcal{P}, \overline{r})$ with bonus $b$:

$$\overline{V}^{\pi}_{h, \mathcal{P}, b}(s, c) := \mathbb{E}_{\pi, \mathcal{P}} \left[ \left( c_h - \sum_{h'=h}^{H} U(r_{h'}) \right)^+ - \sum_{h'=h}^{H} b_{h'}(s_{h'}, a_{h'}) \,\middle|\, s_h = s, c_h = c \right].$$

Similarly, the Q function of $(\mathcal{P}, \overline{r})$ with bonus $b$ is defined as:

$$\overline{Q}^{\pi}_{h, \mathcal{P}, b}(s, c, a) := \mathbb{E}_{\pi, \mathcal{P}} \left[ \left( c_h - \sum_{h'=h}^{H} U(r_{h'}) \right)^+ - \sum_{h'=h}^{H} b_{h'}(s_{h'}, a_{h'}) \,\middle|\, s_h = s, c_h = c, a_h = a \right].$$

To stay consistent with Line 15, we also derive the optimal augmented policy of $(\widehat{\mathcal{P}}, \overline{r})$ with bonus $\widehat{b}$ in Line 16 of Algorithm 1, i.e.,

$$\pi^k \leftarrow \arg \min_{\pi \in \overline{\Pi}_{Aug}} \overline{V}^{\pi}_{1, \widehat{\mathcal{P}}, \widehat{b}}(s_1, c^k).$$

Note that in Line 8 of Algorithm 1 we need to roll out $\pi^k$ in $\mathcal{M}$ to collect samples. Since $\pi^k \in \overline{\Pi}_{Aug}$ works only within the grid, we discretize the reward we observe when executing $\pi^k$ in $\mathcal{M}$, which is equivalent to playing the following augmented policy $\overline{\pi}^k \in \Pi_{Aug}$ in $\mathcal{M}$:

$$\overline{\pi}^k_h \left( s, c^k - \sum_{t=1}^{h-1} r_t \right) = \pi^k_h \left( s, c^k - \sum_{t=1}^{h-1} U(r_t) \right), \forall h \in [H]. \tag{8}$$

Planning within a discretized grid via (7) will inevitably incur errors compared to the original objective. Nevertheless, we can show that if the precision $\upsilon$ is sufficiently small, the discretized MDP $\overline{\mathcal{M}}$ will be an excellent approximation to $\mathcal{M}$ and thus the induced error of planning via (7) will be negligible. Formally, let $\overline{R}(\pi, c)$ denote the return of executing $\pi \in \overline{\Pi}_{Aug}$ with initial budget $c = i\upsilon$ ($0 \le i \le \lceil H/\upsilon \rceil$) in $\overline{\mathcal{M}}$, then we have the following properties from the literature (Wang et al., 2023):

**Proposition B.1.** *For any $0 < \tau < 1, \upsilon > 0$, policy $\pi \in \overline{\Pi}_{Aug}$ and $0 \le i \le \lceil H/\upsilon \rceil$, we have*

(1)  $\mathsf{CVaR}^*_\tau \le \overline{\mathsf{CVaR}}^*_\tau := \underset{\pi \in \overline{\Pi}_{Aug}, 0 \le i \le \lceil H/\upsilon \rceil}{\max} \mathsf{CVaR}_\tau(\overline{R}(\pi, i\upsilon)),$

(2)  $0 \le \mathsf{CVaR}_\tau(\overline{R}(\pi, i\upsilon)) - \mathsf{CVaR}_\tau(R(\overline{\pi}, i\upsilon)) \le \dfrac{H\upsilon}{\tau}.$

## B.2 *CVaR-LSVI*

With discretization, we only need to search within the grid of $c$ in each iteration. For each discrete value $iv$, we apply the *CVaR-LSVI* algorithm as planning oracle as introduced in Section 5. The only difference is that now we simulate with the discretized reward model $\overline{r}$. The full algorithm is shown in Algorithm 2. In particular, in Line 12 we can express $\overline{Q}_h^t$ with $w_h^t$:

$$\overline{Q}_h^t(s, iv, a) = \text{Clip}_{[-H,H]}\left( -\widehat{b}_h(s,a) + \left(\overline{\phi}_h(s,a)\right)^\top w_h^t(iv) - \beta\left\|\overline{\phi}_h(s,a)\right\|_{(\Lambda_h^t)^{-1}} \right). \quad (9)$$

Moreover, note that we can bound the norm of the newly-constructed feature vectors as follows:

$$\left\|\overline{\phi}_h(s,a)\right\|_2 \le 1, \qquad \left\|\sum_{0 \le i \le \lceil 1/v \rceil} \int_{\mathcal{S}} \overline{\psi}_h(s', iv)ds\right\| \le (1 + \lceil 1/v \rceil)\sqrt{d}.$$

This bound will be useful in our analysis.

---

**Algorithm 2** *CVaR-LSVI*

---

**Require:** MDP transition model and reward $(\widehat{\mathcal{P}} = (\widehat{\phi}, \widehat{\psi}), \overline{r})$, bonus $\widehat{b}$, initial budget $i_1 v$, number of iterations $T_1$, parameters $\lambda$ and $\beta$, number of policy evaluation episodes $T_2$.

1: Compute $\overline{\phi}_h(s,a) = \widehat{\phi}_h(s,a) \otimes \phi_{h,r}(s,a)$ for all $h \in [H], s \in \mathcal{S}, a \in \mathcal{A}$ where $\phi_{h,r}(s,a) \in \mathbb{R}^{\lceil 1/v \rceil + 1}$ and $(\phi_{h,r}(s,a))_i = \overline{r}_h(iv|s,a)$ for all $0 \le i \le \lceil 1/v \rceil$.

2: **for** $t = 1, \cdots, T_1$ **do**

3:    Initialize $\overline{V}_{H+1}^t(s, iv) \leftarrow iv$ for all $s \in \mathcal{S}$ and $0 \le i \le \lceil H/v \rceil$.

4:    **for** $h = H, \cdots, 1$ **do**

5:       Compute $\Lambda_h^t \leftarrow \lambda I + \sum_{j=1}^{t-1} \overline{\phi}_h(s_h^j, a_h^j)(\overline{\phi}_h(s_h^j, a_h^j))^\top$.

6:       Calculate $w_h^t(iv) \leftarrow (\Lambda_h^t)^{-1} \sum_{j=1}^{t-1} \overline{\phi}_h(s_h^j, a_h^j) \cdot \left[\overline{V}_{h+1}^t(s_{h+1}^j, iv - \overline{r}_h^j)\right], \forall 0 \le i \le \lceil H/v \rceil$.

7:       **for** $j = 1, \cdots, t-1$ **do**

8:          Compute for all $a \in \mathcal{A}, 0 \le i \le \lceil H/v \rceil$:

$$\overline{Q}_h^t(s_h^j, iv, a) \leftarrow \text{Clip}_{[-H,H]}\left( -\widehat{b}_h(s_h^j, a) + \left(\overline{\phi}_h(s_h^j, a)\right)^\top w_h^t(iv) - \beta\left\|\overline{\phi}_h(s_h^j, a)\right\|_{(\Lambda_h^t)^{-1}} \right).$$

9:          $\overline{V}_h^t(s_h^j, iv) \leftarrow \min_{a \in \mathcal{A}} \overline{Q}_h^t(s_h^j, iv, a)$.

10:       **end for**

11:    **end for**

12:    Simulate the greedy policy $\widetilde{\pi}^t$ (w.r.t. $\overline{Q}_h^t(s, iv, a)$ defined in (9)) with the initial budget $c_1 = i_1 v$ in the MDP $(\widehat{\mathcal{P}}, \overline{r})$ and collect a trajectory $(s_h^t, a_h^t, \overline{r}_h^t)_{h=1}^H$.

13: **end for**

14: **for** $t = 1, \cdots, T_1$ **do**

15:    Simulate $\widetilde{\pi}^t$ with initial budget $c_1 = i_1 v$ in $(\widehat{\mathcal{P}}, \overline{r})$ for $T_2$ episodes and collect trajectories $\left\{\left(s_h^{t,j}, a_h^{t,j}, \overline{r}_h^{t,j}\right)_{h=1}^H\right\}_{j=1}^{T_2}$.

16:    Compute $\widehat{\overline{V}}_1^{\widetilde{\pi}^t}(s_1, i_1 v) \leftarrow \frac{1}{T_2} \sum_{j=1}^{T_2} \left(i_1 v - \sum_{h=1}^H \overline{r}_h^{t,j}(s_h^{t,j}, a_h^{t,j})\right)^+ - \sum_{h=1}^H \widehat{b}_h(s_h^{t,j}, a_h^{t,j})$.

17: **end for**

18: **Return:** value estimate $\min_{t \in [T_1]} \widehat{\overline{V}}_1^{\widetilde{\pi}^t}(s_1, i_1 v)$ and policy $\arg\min_{\widetilde{\pi}^t} \widehat{\overline{V}}_1^{\widetilde{\pi}^t}(s_1, i_1 v)$

---

Now let $\overline{V}_{h,\widehat{\mathcal{P}},\widehat{b}}^*$ denote $\min_{\pi \in \overline{\Pi}_{Aug}} \overline{V}_{h,\widehat{\mathcal{P}},\widehat{b}}^\pi$. Then we have the following theorem indicating that Algorithm 2 can do planning in $(\widehat{\mathcal{P}}, \overline{r})$ accurately with appropriate $T_1$ and $T_2$:

**Theorem B.2.** *Let*

$$\lambda = 1, \beta = \tilde{O}\left(\frac{H^{\frac{3}{2}} d \iota^{\frac{1}{4}}}{v}\right), T_1 = \tilde{O}\left(\frac{H^5 d^3 \iota}{v^3 \varepsilon^2}\right), T_2 = \tilde{O}\left(\frac{H^2 \log \frac{T_1}{\delta}}{\varepsilon^2}\right),$$

*where $\iota = \log^2 \frac{HdT_1}{\upsilon\delta}$, we have with probability at least $1 - \delta$ that*

$$(1) \qquad \widehat{\overline{V}}^*_{1,\widehat{\mathcal{P}},\widehat{b}}(s_1, i_1\upsilon) \leq \overline{V}^*_{1,\widehat{\mathcal{P}},\widehat{b}}(s_1, i_1\upsilon) + \frac{3}{4}\varepsilon,$$

$$(2) \qquad \left| \widehat{\overline{V}}^*_{1,\widehat{\mathcal{P}},\widehat{b}}(s_1, i_1\upsilon) - \overline{V}^{\widehat{\overline{\pi}}}_{1,\widehat{\mathcal{P}},\widehat{b}}(s_1, i_1\upsilon) \right| \leq \frac{1}{4}\varepsilon.$$

*where $\widehat{\overline{\pi}} = \arg\min_{\widetilde{\pi}^t} \widehat{\overline{V}}_1^{\widetilde{\pi}^t}(s_1, i_1\upsilon)$ and $\widehat{\overline{V}}^*_{1,\widehat{\mathcal{P}},\widehat{b}}(s_1, i_1\upsilon) := \min_{\widetilde{\pi}^t} \widehat{\overline{V}}_1^{\widetilde{\pi}^t}(s_1, i_1\upsilon)$ are the returned values of CVaR-LSVI.*

The proof is deferred to Appendix D.1.

## B.3 COMPUTATIONAL COMPLEXITY

Equipping *ELA* with *CVaR-LSVI*, we can derive *ELLA*, which is shown in Algorithm 3. Based on the above discussions about discretization and *CVaR-LSVI*, the computational complexity of *ELLA* for finding an $\epsilon$-policy can be characterized as follows:

**Theorem B.3.** *Let*

$$\alpha^k = O\left(\sqrt{H^2(|\mathcal{A}| + d^2)}\log\left(\frac{|\mathcal{F}|Hk}{\delta}\right)\right), \quad \lambda^k = O\left(d\log\left(\frac{|\mathcal{F}|Hk}{\delta}\right)\right),$$

$$K = \tilde{O}\left(\frac{H^6 A^2 d^4 \log\frac{|\mathcal{F}|}{\delta}}{\tau^2\epsilon^2}\right), \upsilon = \frac{\epsilon\tau}{3H}, \lambda = 1, \beta = \tilde{O}\left(\frac{H^{\frac{3}{2}}d\iota^{\frac{1}{4}}}{\upsilon}\right),$$

$$T_1 = \tilde{O}\left(\frac{H^5 d^3\iota}{\upsilon^3\tau^2\epsilon^2}\right), T_2 = \tilde{O}\left(\frac{H^2\log\frac{T_1}{\delta}}{\tau^2\epsilon^2}\right).$$

*Then we have with probability at least $1 - \delta$,*

$$\mathsf{CVaR}^*_\tau - \mathsf{CVaR}_\tau(R(\widehat{\pi}, \widehat{c})) \leq \epsilon.$$

*In total, the sample complexity is upper bounded by $\tilde{O}\left(\frac{H^7 A^2 d^4 \log\frac{|\mathcal{F}|}{\delta}}{\tau^2\epsilon^2}\right)$. The MLE oracle is called $\tilde{O}\left(\frac{H^7 A^2 d^4 \log\frac{|\mathcal{F}|}{\delta}}{\tau^2\epsilon^2}\right)$ times and the rest of the computation cost is $\tilde{O}\left(\frac{H^{29} A^3 d^{12} \log\frac{|\mathcal{F}|}{\delta}}{\tau^{16}\epsilon^{16}}\right)$.*

The proof is deferred to Appendix D.2. Theorem B.3 indicates that even when the reward is continuous, Algorithm 3 can still achieve polynomial sample complexity and polynomial computational complexity given an MLE oracle.

## C PROOFS FOR SECTION 4

**Proof Sketch** Recall that $\pi^k$ and $c^k$ are the exploration policy and initial budget output from Line 16 of Algorithm 1 at the end of the $k$-th iteration, respectively. In the proof, we show that, with probability at least $1 - \delta$, it holds that

$$\mathrm{Regret}(K) \lesssim \tau^{-1}H^3 Ad^2\sqrt{K}\sqrt{\log\left(1 + \frac{K}{d\lambda^1}\right)\log\left(\frac{|\mathcal{F}|Hk}{\delta}\right)} \tag{10}$$

which is formalized in Lemma C.1. Once it is established, the suboptimality of the uniform mixture of $\{(c^k, \pi^k)\}$ satisfies that

$$\mathsf{CVaR}^*_\tau - \frac{1}{K}\sum_{k=1}^K \mathsf{CVaR}_\tau(R(\pi^k, c^k)) \lesssim \tau^{-1}H^3 Ad^2 K^{-\frac{1}{2}}\sqrt{\log\left(1 + \frac{K}{d\lambda^1}\right)\log\left(\frac{|\mathcal{F}|Hk}{\delta}\right)} \tag{11}$$

Let the RHS smaller than $\epsilon$, we have that when

$$K \gtrsim O\left(\frac{H^6 A^2 d^4}{\tau^2\epsilon^2}\log\left(1 + \frac{K}{d\lambda^1}\right)\log\left(\frac{|\mathcal{F}|Hk}{\delta}\right)\right)$$

---

**Algorithm 3** *ELLA*

---

**Require:** Risk tolerance $\tau \in (0,1]$, number of iterations $K$, parameters $\{\lambda^k\}_{k\in[K]}$ and $\{\alpha^k\}_{k\in[K]}$, models $\mathcal{F} = \{\Psi, \Phi\}$, failure probability $\delta \in (0,1)$, discretization precision $\upsilon$, *CVaR-LSVI* parameters $\lambda, \beta, T_1, T_2$.

1: Set datasets $\mathcal{D}_h, \widetilde{\mathcal{D}}_h \leftarrow \emptyset$ for each $h \in [H-1]$.
2: Calculate the discretized reward distribution $\bar{r}$.
3: Initialize the exploration policy $\pi^0 \leftarrow \{\pi_h^0(s, i\upsilon) = U(\mathcal{A}), \text{for any } s \in \mathcal{S}, 0 \le i \le \lceil H/\upsilon\rceil\}_{h\in[H]}$.
4: Initialize the budget $i^0 \leftarrow \lceil H/\upsilon\rceil$.
5: **for** iteration $k = 1, \dots, K$ **do**
6:     Collect a tuple $(\tilde{s}_1, \tilde{a}_1, s_2')$ by taking $\tilde{a}_1 \sim U(\mathcal{A}), s_2' \sim P_1^*(\cdot|\tilde{s}_1, \tilde{a}_1)$.
7:     Update $\widetilde{\mathcal{D}}_1 \leftarrow \widetilde{\mathcal{D}}_1 \cup \{(\tilde{s}_1, \tilde{a}_1, s_2')\}$.
8:     **for** $h = 1, \cdots, H-1$ **do**
9:       Collect two transition tuples $(s_h, a_h, \tilde{s}_{h+1})$ and $(\tilde{s}_{h+1}, \tilde{a}_{h+1}, s_{h+2}')$ by first rolling out $\bar{\pi}^{k-1}$ (defined in (8)) starting from $(s_1, i^{k-1}\upsilon)$ into state $s_h$, taking $a_h \sim U(\mathcal{A})$, and receiving $\tilde{s}_{h+1} \sim P_h^*(\cdot|s_h, a_h)$, then taking $\tilde{a}_{h+1} \sim U(\mathcal{A})$ and receiving $s_{h+2}' \sim P_{h+1}^*(\cdot|\tilde{s}_{h+1}, \tilde{a}_{h+1})$.
10:       Update $\mathcal{D}_h \leftarrow \mathcal{D}_h \cup \{(s_h, a_h, \tilde{s}_{h+1})\}$.
11:       Update $\widetilde{\mathcal{D}}_{h+1} \leftarrow \widetilde{\mathcal{D}}_{h+1} \cup \{(\tilde{s}_{h+1}, \tilde{a}_{h+1}, s_{h+2}')\}$ if $h \le H-2$.
12:       Learn representations via MLE

$$\widehat{P}_h := (\widehat{\psi}_h, \widehat{\phi}_h) \leftarrow \arg\max_{(\psi,\phi)\in\mathcal{F}} \sum_{(s_h, a_h, s_{h+1})\in\{\mathcal{D}_h + \widetilde{\mathcal{D}}_h\}} \log\langle\psi(s_{h+1}), \phi(s_h, a_h)\rangle$$

13:       Update empirical covariance matrix $\widehat{\Sigma}_h = \sum_{(s,a)\in\mathcal{D}_h} \widehat{\phi}_h(s,a)\widehat{\phi}_h(s,a)^\top + \lambda^k I_d$.
14:       Set the exploration bonus:

$$\widehat{b}_h(s,a) \leftarrow \begin{cases} \min\left(\alpha^k\sqrt{\widehat{\phi}_h(s,a)\widehat{\Sigma}_h^{-1}\widehat{\phi}_h(s,a)^\top}, 2\right) & h \le H-2 \\ 0 & h = H-1 \end{cases}$$

15:     **end for**
16:     **for** $i = 0, 1, \cdots, \lceil H/\upsilon\rceil$ **do**
17:       Run *CVaR-LSVI* (Algorithm 2) with MDP model $(\widehat{P}, \bar{r})$, bonus $\widehat{b}$, initial budget $i\upsilon$ and parameters $(\lambda, \beta, T_1, T_2)$ and let the returned value estimate and policy be $\widehat{\widetilde{V}}_1^*(s_1, i\upsilon)$ and $\widehat{\widetilde{\pi}}(i)$.
18:     **end for**
19:     Obtain $i^k \leftarrow \arg\max_{0\le i\le\lceil H/\upsilon\rceil}\left\{i\upsilon - \tau^{-1}\widehat{\widetilde{V}}_1^*(s_1, i\upsilon)\right\}$ and $\pi^k \leftarrow \widehat{\widetilde{\pi}}(i^k)$.
20: **end for**
**Ensure:** Uniformly sample $k$ from $[K]$, return $(\widehat{\pi}, \widehat{c}) = (\bar{\pi}^k, i^k\upsilon)$.

---

the uniform mixture of $\{(\pi^k, c^k)\}$ is an $\epsilon$-optimal policy. Finally, noting that $H$ trajectories are collected per iteration, we conclude the proof of Theorem 4.1. To establish (10), we decompose the suboptimality of the $k$-th iteration into

$$\mathsf{CVaR}_\tau^* - \mathsf{CVaR}_\tau(R(\pi^k, c^k))$$

$$= c^* - \tau^{-1}V_{1,\widehat{\mathcal{P}}^k, \widehat{b}^k}^{\pi^*}(s_1, c^*) - \mathsf{CVaR}_\tau(R(\pi^k, c^k)) + \mathsf{CVaR}_\tau(R(\pi^*, c^*)) - \left(c^* - \tau^{-1}V_{1,\widehat{\mathcal{P}}^k, \widehat{b}^k}^{\pi^*}(s_1, c^*)\right)$$

$$\le c^k - \tau^{-1}V_{1,\widehat{\mathcal{P}}^k, \widehat{b}^k}^{\pi^k}(s_1, c^k) - \mathsf{CVaR}_\tau(R(\pi^k, c^k)) + c^* - \tau^{-1}V_{1,\mathcal{P}^*, 0}^{\pi^*}(s_1, c^*) - \left(c^* - \tau^{-1}V_{1,\widehat{\mathcal{P}}^k, \widehat{b}^k}^{\pi^*}(s_1, c^*)\right)$$

$$\le \tau^{-1}\underbrace{\left(V_{1,\mathcal{P}^*, 0}^{\pi^k}(s_1, c^k) - V_{1,\widehat{\mathcal{P}}^k, \widehat{b}^k}^{\pi^k}(s_1, c^k)\right)}_{\text{(i)}} + \tau^{-1}\underbrace{\left(V_{1,\widehat{\mathcal{P}}^k, \widehat{b}^k}^{\pi^*}(s_1, c^*) - V_{1,\mathcal{P}^*, 0}^{\pi^*}(s_1, c^*)\right)}_{\text{(ii)}}$$

where the first inequality holds by the fact that $\pi^k$ is greedy (Line 16 of Algorithm 1) and the last inequality holds by

$$\mathsf{CVaR}_\tau(R(\pi^k, c^k)) = \sup_{t \in \mathbb{R}} \left( t - \tau^{-1} \mathbb{E}_{R \sim R(\pi^k, c^k)}[(t - R)^+] \right) \geq c^k - \tau^{-1} \mathbb{E}_{R \sim R(\pi^k, c^k)}[(c^k - R)^+]$$

Utilizing simulation lemma C.3 for risk-sensitive RL, we further upper bound terms (i) and (ii) by the error in estimating the transition kernel and the bonus terms, i.e.,

$$\text{term (i)} \leq \mathbb{E}_{(s,a) \sim d_h^{(\pi^k, c^k)}} \left[ \widehat{b}_h^k(s, a) \right] + 3H \cdot \mathbb{E}_{(s,a) \sim d_h^{(\pi^k, c^k)}} \left[ f_h^k(s, a) \right]$$

$$\text{term (ii)} \leq \mathbb{E}_{(s,a) \sim d_{h, \widehat{\mathcal{P}}^k}^{(\pi^*, c^*)}} \left[ H \cdot f_h^k(s, a) - \widehat{b}_h^k(s, a) \right]$$

where $f_h^k(s, a) := \|\widehat{P}_h^k(\cdot|s, a) - P_h^*(\cdot|s, a)\|_1$. By the analysis in Lemma C.1 and C.2, we prove the inequality (10).

Before proceeding to the detailed proofs, we first introduce the essential regret decomposition.

**Lemma C.1** (Regret). *With probability $1 - \delta$, we have that*

$$\sum_{k=1}^K \mathsf{CVaR}_\tau^* - \mathsf{CVaR}_\tau(R(\pi^k, c^k)) \lesssim \tau^{-1} H^3 A d^2 \sqrt{K} \sqrt{\log\left(1 + \frac{K}{d\lambda^1}\right) \log\left(\frac{|\mathcal{F}|Hk}{\delta}\right)} \quad (12)$$

*Proof.* Letting $f_h^k(s, a) = \|\widehat{P}_h^k(\cdot|s, a) - P_h^*(\cdot|s, a)\|$, we condition on the event that for all $(k, h) \in [K] \times [H]$, the following inequalities hold

$$\mathbb{E}_{s \sim \rho_h^k, a \sim U(\mathcal{A})} \left[ \left( f_h^k(s, a) \right)^2 \right] \leq \zeta^k, \ \mathbb{E}_{s \sim \eta_h^k, a \sim U(\mathcal{A})} \left[ \left( f_h^k(s, a) \right)^2 \right] \leq \zeta^k$$

$$\|\phi(s, a)\|_{(\widehat{\Sigma}_{h, \phi}^k)^{-1}} = \Theta(\|\phi(s, a)\|_{(\Sigma_{\rho_h^k \times U(\mathcal{A}), \phi})^{-1}})$$

By Lemmas E.1 and E.2, this event happens with probability $1 - \delta$. For any iteration $k$, we have that For a fixed iteration $k$, we have that

$$\mathsf{CVaR}_\tau^* - \mathsf{CVaR}_\tau(R(\pi^k, c^k))$$

$$= c^* - \tau^{-1} V_{1, \widehat{\mathcal{P}}^k, \widehat{b}^k}^{\pi^*}(s_1, c^*) - \mathsf{CVaR}_\tau(R(\pi^k, c^k)) + \mathsf{CVaR}_\tau(R(\pi^*, c^*)) - \left( c^* - \tau^{-1} V_{1, \widehat{\mathcal{P}}^k, \widehat{b}^k}^{\pi^*}(s_1, c^*) \right)$$

$$\leq c^k - \tau^{-1} V_{1, \widehat{\mathcal{P}}^k, \widehat{b}^k}^{\pi^k}(s_1, c^k) - \mathsf{CVaR}_\tau(R(\pi^k, c^k)) + c^* - \tau^{-1} V_{1, \mathcal{P}^*, 0}^{\pi^*}(s_1, c^*) - \left( c^* - \tau^{-1} V_{1, \widehat{\mathcal{P}}^k, \widehat{b}^k}^{\pi^*}(s_1, c^*) \right)$$

$$\leq \tau^{-1} \left( V_{1, \mathcal{P}^*, 0}^{\pi^k}(s_1, c^k) - V_{1, \widehat{\mathcal{P}}^k, \widehat{b}^k}^{\pi^k}(s_1, c^k) \right) + \tau^{-1} \underbrace{\left( V_{1, \widehat{\mathcal{P}}^k, \widehat{b}^k}^{\pi^*}(s_1, c^*) - V_{1, \mathcal{P}^*, 0}^{\pi^*}(s_1, c^*) \right)}_{\leq \sqrt{H^2 A \zeta^k} \text{ by Lemma C.2}} \quad (13)$$

where the first inequality holds by the fact that $\pi^k$ is greedy (Line 15 of Algorithm 1) and the last inequality holds by

$$\mathsf{CVaR}_\tau(R(\pi^k, c^k)) = \sup_{t \in \mathbb{R}} \left( t - \tau^{-1} \mathbb{E}_{R \sim R(\pi^k, c^k)}[(t - R)^+] \right) \geq c^k - \tau^{-1} \mathbb{E}_{R \sim R(\pi^k, c^k)}[(c^k - R)^+]$$

Therefore, it remains to bound the first term, which by simulation lemma C.3, can be further written as

$$V_{1, \mathcal{P}^*, 0}^{\pi^k}(s_1, c^k) - V_{1, \widehat{\mathcal{P}}^k, \widehat{b}^k}^{\pi^k}(s_1, c^k)$$

$$= \mathbb{E}_{\pi^k, \mathcal{P}^*} \left[ \left( c^k - \sum_{h=1}^H r_h(s_h, a_h) \right)^+ \right] - \mathbb{E}_{\pi^k, \widehat{\mathcal{P}}^k} \left[ \left( c^k - \sum_{h=1}^H r_h(s_h, a_h) \right)^+ - \sum_{h=1}^H \widehat{b}_h^k(s_h, a_h) \Big| c_1 = c^k \right]$$

$$\leq \sum_{h=1}^H \mathbb{E}_{\pi^k, \widehat{\mathcal{P}}^k} \left[ \widehat{b}_h^k(s_h, a_h) \Big| c_1 = c^k \right] + H \cdot \sum_{h=1}^H \mathbb{E}_{(s,a) \sim d_h^{(\pi^k, c^k)}} \left[ f_h^k(s, a) \right]$$

$$\leq \underbrace{\sum_{h=1}^H \mathbb{E}_{(s,a) \sim d_h^{(\pi^k, c^k)}} \left[ \widehat{b}_h^k(s, a) \right]}_{\text{(i)}} + 3H \cdot \underbrace{\sum_{h=1}^H \mathbb{E}_{(s,a) \sim d_h^{(\pi^k, c^k)}} \left[ f_h^k(s, a) \right]}_{\text{(ii)}} \quad (14)$$

where the last inequality holds by the simulation Lemma E.5 for risk-neutral RL and the fact that $\|\widehat{b}_h^k\|_\infty \le 2$. where the last inequality holds by $\|V_{h,\widehat{P}^k,r+\widehat{b}^k}^{\pi^k}\|_\infty \le 2$. We first consider term (i). By Lemma E.4 and noting that the bonus term $\widehat{b}_h^k$ is $O(1)$, we have

$$\sum_{h=1}^{H-1} \mathbb{E}_{(s,a)\sim d_h^{(\pi^k,c^k)}} \left[\widehat{b}_h^k(s,a)\right]$$

$$\lesssim \sum_{h=1}^{H-1} \mathbb{E}_{(s,a)\sim d_h^{(\pi^k,c^k)}} \left[\min\left\{\alpha^k\|\widehat{\phi}_h^k(s,a)\|_{\Sigma_{\rho_h^k\times U(\mathcal{A}),\widehat{\phi}_h^k}^{-1}}, 2\right\}\right]$$

$$\le \sqrt{A\cdot(\alpha^k)^2\cdot\mathbb{E}_{s\sim\rho_1^k,a\sim U(\mathcal{A})}\left[\|\widehat{\phi}_1^k(s,a)\|_{\Sigma_{\rho_1^k\times U(\mathcal{A}),\widehat{\phi}_1^k}^{-1}}^2\right]}$$

$$+ \sum_{h=1}^{H-2} \mathbb{E}_{(s,a)\sim d_{h+1}^{(\pi^k,c^k)}} \left[\alpha^k\|\widehat{\phi}_{h+1}^k(s,a)\|_{\Sigma_{\rho_{h+1}^k\times U(\mathcal{A}),\widehat{\phi}_{h+1}^k}^{-1}}\right]$$

$$\le \sum_{h=1}^{H-2} \mathbb{E}_{(s,a)\sim d_h^{(\pi^k,c^k)}} \left[\|\phi_h^*(s,a)\|_{\Sigma_{\rho_h^k,\phi^*}^{-1}}\sqrt{k(\alpha^k)^2 A\cdot\mathbb{E}_{s\sim\rho_{h+1}^k,a\sim U(\mathcal{A})}\left[\|\widehat{\phi}_{h+1}^k(s,a)\|_{\Sigma_{\rho_{h+1}^k\times U(\mathcal{A}),\widehat{\phi}_{h+1}^k}^{-1}}^2\right]+4\lambda^k d}\right]$$

$$+ \sqrt{A\cdot(\alpha^k)^2\cdot\mathbb{E}_{s\sim\rho_1^k,a\sim U(\mathcal{A})}\left[\|\widehat{\phi}_1^k(s,a)\|_{\Sigma_{\rho_1^k\times U(\mathcal{A}),\widehat{\phi}_1^k}^{-1}}^2\right]}$$

$$\le \sqrt{\frac{dA(\alpha^k)^2}{k}} + \sqrt{dA(\alpha^k)^2+4\lambda^k d}\cdot\sum_{h=1}^{H-2}\mathbb{E}_{(s,a)\sim d_h^{(\pi^k,c^k)}}\left[\|\phi_h^*(s,a)\|_{\Sigma_{\rho_h^k,\phi^*}^{-1}}\right] \tag{15}$$

where the last inequality holds by

$$k\cdot\mathbb{E}_{s\sim\rho_h^k,a\sim U(\mathcal{A})}\left[\|\widehat{\phi}_h^k(s,a)\|_{\Sigma_{\rho_h^k\times U(\mathcal{A}),\widehat{\phi}_h^k}^{-1}}^2\right] = k\mathrm{Tr}\left(\mathbb{E}_{\rho_h^k\times U(\mathcal{A})}[\widehat{\phi}_h^k(\widehat{\phi}_h^k)^\top]\left\{k\mathbb{E}_{\rho_h^k\times U(\mathcal{A})}[\widehat{\phi}_h^k(\widehat{\phi}_h^k)^\top]+\lambda^k\right\}^{-1}\right)\le d$$

Similarly, for term (ii), we have that

$$\sum_{h=1}^{H-1} \mathbb{E}_{(s,a)\sim d_h^{(\pi^k,c^k)}} \left[f_h^k(s,a)\right]$$

$$\le \sqrt{A\cdot\mathbb{E}_{s\sim\rho_1^k,a\sim U(\mathcal{A})}\left[\left(f_1^k(s,a)\right)^2\right]}$$

$$+ \sum_{h=1}^{H-2} \mathbb{E}_{(s,a)\sim d_h^{(\pi^k,c^k)}} \left[\|\phi_h^*(s,a)\|_{\Sigma_{\rho_h^k,\phi^*}^{-1}}\sqrt{kA\cdot\mathbb{E}_{s\sim\rho_{h+1}^k,a\sim U(\mathcal{A})}\left[\left(f_{h+1}^k(s,a)\right)^2\right]+4\lambda^k d}\right]$$

$$\le \sqrt{A\zeta_1^k} + \alpha^k\cdot\sum_{h=1}^{H-2}\mathbb{E}_{(s,a)\sim d_h^{(\pi^k,c^k)}}\left[\|\phi_h^*(s,a)\|_{\Sigma_{\rho_h^k,\phi^*}^{-1}}\right], \tag{16}$$

where the last step uses

$$\alpha^k \lesssim \sqrt{H^2 A d^2\log\left(\frac{|\mathcal{F}|Hk}{\delta}\right)}.$$

Combining (13), (14), (15), and (16) we obtain

$$\tau\left(\sum_{k=1}^K\left(\mathsf{CVaR}_\tau^* - \mathsf{CVaR}_\tau(R(\pi^k,c^k))\right)\right)$$

$$\lesssim \sum_{k=1}^K\left(\sqrt{\frac{dA(\alpha^k)^2}{k}}+\sqrt{H^2A\zeta^k}\right)+\sum_{k=1}^K\sum_{h=1}^{H-2}\left(2H\cdot\alpha^k\cdot\mathbb{E}_{(s,a)\sim d_h^{(\pi^k,c^k)}}\left[\|\phi_h^*(s,a)\|_{\Sigma_{\rho_h^k,\phi^*}^{-1}}\right]\right)$$

$$+ \sum_{k=1}^{K} \sqrt{dA(\alpha^k)^2 + 4\lambda^k d} \sum_{h=1}^{H-2} \mathbb{E}_{(s,a)\sim d_h^{(\pi^k,c^k)}} \left[ \|\phi_h^*(s,a)\|_{\Sigma_{\rho_h^k,\phi^*}^{-1}} \right]$$

$$\lesssim H^2 A d^{\frac{3}{2}} \sqrt{\log\left(\frac{|\mathcal{F}|Hk}{\delta}\right)} \sum_{k=1}^{K}\sum_{h=1}^{H-2} \mathbb{E}_{(s,a)\sim d_h^{(\pi^k,c^k)}} \left[ \|\phi_h^*(s,a)\|_{\Sigma_{\rho_h^k,\phi^*}^{-1}} \right]$$

$$\lesssim H^3 A d^2 \sqrt{K} \sqrt{\log\left(1 + \frac{K}{d\lambda^1}\right) \log\left(\frac{|\mathcal{F}|Hk}{\delta}\right)}$$

where the last inequality holds by (Uehara et al., 2022, Lemma 18), i.e.,

$$\sum_{k=1}^{K} \mathbb{E}_{(s,a)\sim d_h^{(\pi^k,c^k)}} \left[ \|\phi_h^*(s,a)\|_{\Sigma_{\rho_h^k,\phi^*}^{-1}} \right] \leq \sqrt{dK\log\left(1 + \frac{K}{d\lambda^1}\right)}$$

Therefore, we conclude the proof. $\qquad\square$

**Lemma C.2** (Almost Optimism at the Initial Distribution). *Consider an episode $k \in [K]$ and set*

$$\alpha^k = \sqrt{H^2(A+d^2)\log\left(\frac{|\mathcal{F}|Hk}{\delta}\right)}, \lambda^k = O\left(d\log\left(\frac{|\mathcal{F}|Hk}{\delta}\right)\right), \zeta^k = O\left(\frac{1}{k}\log\left(\frac{|\mathcal{F}|Hk}{\delta}\right)\right)$$
(17)

*with probability $1-\delta$, we have that*

$$V_{1,\widehat{\mathcal{P}}^k,\widehat{b}^k}^{\pi^*}(s_1,c^*) - V_{1,\mathcal{P}^*,0}^{\pi^*}(s_1,c^*) \leq \sqrt{H^2 A \zeta^k}$$
(18)

*Proof.* Similar to the proof of Lemma C.1, letting $f_h^k(s,a) = \|\widehat{P}_h^k(\cdot|s,a) - P_h^*(\cdot|s,a)\|_1$, we condition on the event that for all $(k,h) \in [K] \times [H]$, the following inequalities hold

$$\mathbb{E}_{s\sim\rho_h^k,a\sim U(\mathcal{A})}\left[\left(f_h^k(s,a)\right)^2\right] \leq \zeta^k, \ \mathbb{E}_{s\sim\eta_h^k,a\sim U(\mathcal{A})}\left[\left(f_h^k(s,a)\right)^2\right] \leq \zeta^k$$
$$\|\phi(s,a)\|_{(\widehat{\Sigma}_{h,\phi}^k)^{-1}} = \Theta(\|\phi(s,a)\|_{(\Sigma_{\rho_h^k \times U(\mathcal{A}),\phi})^{-1}})$$

From Lemmas E.1 and E.2, this event happens with probability $1-\delta$. Note that $b_H^k(s,a) = f_H^k(s,a) := 0$ for any $(s,a) \in \mathcal{S} \times \mathcal{A}$. Then, for any policy $\pi$, from the simulation Lemma C.3, we have that

$$V_{1,\widehat{\mathcal{P}}^k,\widehat{b}^k}^{\pi^*}(s_1,c^*) - V_{1,\mathcal{P}^*,0}^{\pi^*}(s_1,c^*)$$

$$= \mathbb{E}_{\pi^*,\widehat{\mathcal{P}}^k}\left[\left(c^* - \sum_{h=1}^{H} r_h(s_h,a_h)\right)^+ - \sum_{h=1}^{H}\widehat{b}_h^k(s_h,a_h)\Bigg| c_1 = c^*\right] - \mathbb{E}_{\pi^*,\mathcal{P}^*}\left[\left(c^* - \sum_{h=1}^{H} r_h(s_h,a_h)\right)^+\right]$$

$$\leq \sum_{h=1}^{H-1} \mathbb{E}_{(s,a)\sim d_{h,\widehat{\mathcal{P}}^k}^{(\pi^*,c^*)}}\left[H \cdot f_h^k(s,a) - \widehat{b}_h^k(s,a)\right]$$
(19)

For any $h+1 \in \{2,\cdots,H-1\}$, by Lemma E.3 and noting that $\|f_{h+1}\|_\infty \leq 2$, we have that

$$\left|\mathbb{E}_{(s',a')\sim d_{h+1,\widehat{\mathcal{P}}^k}^{(\pi^*,c^*)}}[f_{h+1}^k(s',a')]\right|$$

$$\leq \mathbb{E}_{(s,a)\sim d_{h,\widehat{\mathcal{P}}^k}^{(\pi^*,c^*)}}\left[\|\widehat{\phi}_h^k(s,a)\|_{\Sigma_{\rho_h^k \times U(\mathcal{A}),\widehat{\phi}_h^k}^{-1}} \cdot \sqrt{kA \cdot \mathbb{E}_{s'\sim\eta_{h+1}^k,a'\sim U(\mathcal{A})}\left[\left(f_{h+1}^k(s',a')\right)^2\right] + 4\lambda^k d + 4k\zeta^k}\right]$$

Hence,

$$-\mathbb{E}_{(s',a')\sim d_{h+1,\widehat{\mathcal{P}}^k}^{(\pi^*,c^*)}}[f_{h+1}^k(s',a')] \geq -\sqrt{\beta^k} \cdot \mathbb{E}_{(s,a)\sim d_{h,\widehat{\mathcal{P}}^k}^{(\pi^*,c^*)}}\left[\|\widehat{\phi}_h^k(s,a)\|_{\Sigma_{\rho_h^k \times U(\mathcal{A}),\widehat{\phi}_h^k}^{-1}}\right]$$
(20)

where

$$\beta^k := kA\zeta^k + \lambda^k d + k\zeta^k \lesssim (A+d^2)\log(|\mathcal{F}|Hk/\delta)$$

Combining (19) and (20), we further derive

$$V_{1,\widehat{\mathcal{P}}^k,\widehat{b}^k}^{\pi^*}(s_1,c^*) - V_{1,\mathcal{P}^*,0}^{\pi^*}(s_1,c^*)$$

$$\leq -\sum_{h=1}^{H-1}\mathbb{E}_{(s,a)\sim d_{h,\widehat{\mathcal{P}}^k}^{(\pi^*,c^*)}}\left[\widehat{b}_h^k(s,a)\right] + \sqrt{H^2 A\zeta^k} + H\cdot\sum_{i=1}^{H-2}\mathbb{E}_{(s,a)\sim d_{i+1,\widehat{\mathcal{P}}^k}^{(\pi^*,c^*)}}\left[f_{i+1}^k(s,a)\right]$$

$$\leq -\sum_{h=1}^{H-1}\mathbb{E}_{(s,a)\sim d_{h,\widehat{\mathcal{P}}^k}^{(\pi^*,c^*)}}\left[\widehat{b}_h^k(s,a)\right] + \sum_{h=1}^{H-2}\mathbb{E}_{(s,a)\sim d_{h,\widehat{\mathcal{P}}^k}^{(\pi^*,c^*)}}\left[\min\left\{\alpha^k\|\widehat{\phi}_h^k(s,a)\|_{\Sigma_{\rho_h^k\times U(\mathcal{A}),\widehat{\phi}_h^k}^{-1}},2\right\}\right] + \sqrt{H^2 A\zeta^k}$$

$$\lesssim \sqrt{H^2 A\zeta^k}$$

Therefore, we conclude the proof. $\qquad\square$

**Lemma C.3** (Simulation lemma for risk-sensitive RL). *Given two episodic MDPs* $(H,\mathcal{P} = \{P_h\}_{h\in[H]},r)$ *and* $(H,\widehat{\mathcal{P}} = \{\widehat{P}_h\}_{h\in[H]},r)$, *for any fixed* $c \in [0,1]$ *and policy* $\pi = \{\pi_h : \mathcal{S}\times[0,H]\mapsto\Delta(\mathcal{A})\}_{h\in[H]}$, *we have that*

$$V_{1,\mathcal{P},0}^{\pi}(s_1,c) - V_{1,\widehat{\mathcal{P}},0}^{\pi}(s_1,c) \leq H\cdot\sum_{h=1}^{H}\mathbb{E}_{(s,a)\sim d_{h,\mathcal{P}}^{(\pi,c)}}\left[f_h(s,a)\right] \qquad(21)$$

*where* $f_h(s,a) := \|P_h(\cdot|s,a) - \widehat{P}_h(\cdot|s,a)\|_1$ *for any* $(h,s,a)\in[H]\times\mathcal{S}\times\mathcal{A}$.

*Proof.* By definition, we derive that

$$V_{1,\mathcal{P},0}^{\pi}(s_1,c) - V_{1,\widehat{\mathcal{P}},0}^{\pi}(s_1,c)$$

$$=\mathbb{E}_{a_1\sim\pi_1(\cdot|s_1,c),r_1}\left\{\mathbb{E}_{s'\sim P_1(\cdot|s_1,a_1)}\left[V_{2,\mathcal{P},0}^{\pi}(s',c-r_1)\right] - \mathbb{E}_{s'\sim\widehat{P}_1(\cdot|s_1,a_1)}\left[V_{2,\widehat{\mathcal{P}},0}^{\pi}(s',c-r_1)\right]\right\}$$

$$=\mathbb{E}_{a_1\sim\pi_1(\cdot|s_1,c),r_1}\left\{\mathbb{E}_{s'\sim\widehat{P}_1(\cdot|s_1,a_1)}\left[V_{2,\mathcal{P},0}^{\pi}(s',c-r_1) - V_{2,\widehat{\mathcal{P}},0}^{\pi}(s',c-r_1)\right]\right.$$

$$\left. + \mathbb{E}_{s'\sim P_1(\cdot|s_1,a_1)}\left[V_{2,\mathcal{P},0}^{\pi}(s',c-r_1)\right] - \mathbb{E}_{s'\sim\widehat{P}_1(\cdot|s_1,a_1)}\left[V_{2,\mathcal{P},0}^{\pi}(s',c-r_1)\right]\right\}$$

$$\leq\mathbb{E}_{a_1\sim\pi_1(\cdot|s_1,c),r_1}\mathbb{E}_{s'\sim\widehat{P}_1(\cdot|s_1,a_1)}\left[V_{2,\mathcal{P},0}^{\pi}(s',c-r_1) - V_{2,\widehat{\mathcal{P}},0}^{\pi}(s',c-r_1)\right] + H\cdot\mathbb{E}_{(s,a)\sim d_{1,\mathcal{P}}^{\pi}}[f_1(s,a)]$$

$$\leq\cdots\leq H\cdot\sum_{h=1}^{H}\mathbb{E}_{(s,a)\sim d_{h,\mathcal{P}}^{(\pi,c)}}[f_h(s,a)]$$

which concludes the proof. $\qquad\square$

# D PROOFS FOR APPENDIX B

## D.1 PROOF OF THEOREM B.2

In the following discussion we use $\overline{\phi}_h^j$ to denote $\overline{\phi}_h(s_h^j,a_h^j)$ and $(\widehat{\mathcal{P}}_h\overline{V})(s,iv,a)$ to denote

$$\mathbb{E}_{r\sim\overline{r}_h(\cdot|s,a),s'\sim\widehat{P}_h(\cdot|s,a)}[\overline{V}(s',iv-r)].$$

We also use $\mathcal{G}_{t,h}$ denote the filtration generated by $\{(s_{h'}^j,a_{h'}^j,\overline{r}_{h'}^j)_{h'=1}^H\}_{j=1}^{t-1}\cup(s_{h'}^t,a_{h'}^t,\overline{r}_{h'}^t)_{h'=1}^{h-1}\cup(s_h^t,a_h^t)$.

First, note that we have the following concentration lemma:

**Lemma D.1.** *For all* $h\in[H],t\in[T_1],0\leq i\leq\lceil H/v\rceil$, *with probability at* $1-\delta/4$, *we have*

$$\left\|\sum_{j=1}^{t-1}\overline{\phi}_h^j\left[\overline{V}_{h+1}^t(s_{h+1}^j,iv-\overline{r}_h^j) - \widehat{\mathcal{P}}_h\overline{V}_{h+1}^t(s_h^j,iv,a_h^j)\right]\right\|_{(\Lambda_h^t)^{-1}} \leq\tilde{O}\left(\frac{H^{\frac{3}{2}}d}{v}\sqrt{\log\frac{HdT_1}{v\delta}}\right).$$

The proof is deferred to Appendix D.3. Let $\mathcal{E}_1$ denote the event in Lemma D.1.

On the other hand, we can further bound the difference between $\overline{Q}_h^\pi(s, iv, a)$ and $-\widehat{b}_h(s, a) + \left(\overline{\phi}_h(s, a)\right)^\top w_h^t(iv)$ as follows:

**Lemma D.2.** *For any policy $\pi$, conditioned on $\mathcal{E}_1$, we have for all $s \in \mathcal{S}, 0 \leq i \leq \lceil H/v \rceil, a \in \mathcal{A}, h \in [H], t \in [T_1]$ that*

$$-\widehat{b}_h(s, a) + \left(\overline{\phi}_h(s, a)\right)^\top w_h^t(iv) - \overline{Q}_h^\pi(s, iv, a) = \left(\widehat{\mathcal{P}}_h\left(\overline{V}_{h+1}^t - \overline{V}_{h+1}^\pi\right)\right)(s, iv, a) + \xi_h^t(s, iv, a),$$

*where $|\xi_h^t(s, iv, a)| \leq \beta \left\|\overline{\phi}_h(s, a)\right\|_{(\Lambda_h^t)^{-1}}$.*

The proof of Lemma D.2 follows the same arguments in the proof of Jin et al. (2020)[Lemma B.4] and thus is omitted here.

With Lemma D.2, we can prove that the estimated Q function $\overline{Q}^t$ is optimistic:

**Lemma D.3.** *Conditioned on event $\mathcal{E}_1$, we have for all $s \in \mathcal{S}, 0 \leq i \leq \lceil H/v \rceil, h \in [H], t \in [T_1]$ that $\overline{V}_h^t(s, iv) \leq \overline{V}_h^*(s, iv)$ where $\overline{V}_h^*(s, iv) = \sup_\pi \overline{V}_h^\pi(s, iv)$.*

The proof is deferred to Appendix D.4.

Combining Lemma D.2 and Lemma D.3, we can bound the regret of Algorithm 2 as follows:

**Lemma D.4.** *With probability at least $1 - \delta/2$, we have*

$$\sum_{t=1}^{T_1} \overline{V}_1^{\widetilde{\pi}^t}(s_1, i_1 v) - \overline{V}_1^*(s_1, i_1 v) \leq \tilde{O}\left(\sqrt{\frac{H^5 d^3 T_1 \iota}{v^3}}\right),$$

*where $\iota = \log^2 \frac{H d T_1}{v \delta}$.*

The proof is deferred to Appendix D.5. Denote the event in Lemma D.4 by $\mathcal{E}_2$.

Lemma D.4 implies that by setting $T_1 = \tilde{O}\left(\frac{H^5 d^3 \iota}{v^3 \varepsilon^2}\right)$, we have conditioned on event $\mathcal{E}_2$ that

$$\min_{t \in [T_1]} \overline{V}_1^{\widetilde{\pi}^t}(s_1, i_1 v) - \overline{V}_1^*(s_1, i_1 v) \leq \varepsilon/2.$$

Let $t_1$ denote $\arg \min_{t \in [T_1]} \overline{V}_1^{\widetilde{\pi}^t}(s_1, i_1 v)$. Then on the other hand, by setting $T_2 = \tilde{O}\left(\frac{H^2 \log \frac{T_1}{\delta}}{\varepsilon^2}\right)$, with probability at least $1 - \delta/2$ we have that for all $t \in [T_1]$,

$$\left|\widehat{\overline{V}}_1^{\widetilde{\pi}^t}(s_1, i_1 v) - \overline{V}_1^{\widetilde{\pi}^t}(s_1, i_1 v)\right| \leq \varepsilon/4.$$

Denote the above event by $\mathcal{E}_3$ and let $\widehat{\widetilde{\pi}}$ denote $\arg \min_{\widetilde{\pi}^t} \widehat{\overline{V}}_1^{\widetilde{\pi}^t}(s_1, i_1 v)$. Then conditioned on event $\mathcal{E}_2 \cap \mathcal{E}_3$, we have

$$\widehat{\overline{V}}_1^{\widehat{\widetilde{\pi}}}(s_1, i_1 v) - \overline{V}_1^*(s_1, i_1 v) \leq \widehat{\overline{V}}_1^{\widetilde{\pi}^{t_1}}(s_1, i_1 v) - \overline{V}_1^*(s_1, i_1 v) \leq \overline{V}_1^{\widetilde{\pi}^{t_1}}(s_1, i_1 v) - \overline{V}_1^*(s_1, i_1 v) + \frac{\varepsilon}{4} \leq \frac{3}{4}\varepsilon.$$

## D.2 PROOF OF THEOREM B.3

First, we show that Algorithm 3 can achieve sublinear regret in the discretized MDP $\overline{\mathcal{M}}$. Let $(\pi_{\mathsf{dis}}^*, i^*) := \arg \max_{\pi \in \overline{\Pi}_{Aug}, 0 \leq i \leq \lceil H/v \rceil} \mathsf{CVaR}_\tau(\overline{R}(\pi, iv))$. Note that from (4) we have

$$\overline{\mathsf{CVaR}}_\tau^* = i^* v - \tau^{-1} \overline{V}_{1, \mathcal{P}^*, 0}^{\pi_{\mathsf{dis}}^*}(s_1, i^* v).$$

This implies that

$$\overline{\mathsf{CVaR}}_\tau^* - \mathsf{CVaR}_\tau(\overline{R}(\pi^k, i^k v))$$

$$= \left( i^* \upsilon - \tau^{-1} \overline{V}_{1, \widehat{\mathcal{P}}^k, \widehat{b}^k}^{\pi_{\mathrm{dis}}^*}(s_1, i^* \upsilon) \right) - \mathsf{CVaR}_\tau(\overline{R}(\pi^k, i^k \upsilon)) + \left( i^* \upsilon - \tau^{-1} \overline{V}_{1, \mathcal{P}^*, 0}^{\pi_{\mathrm{dis}}^*}(s_1, i^* \upsilon) \right)$$

$$- \left( i^* \upsilon - \tau^{-1} \overline{V}_{1, \widehat{\mathcal{P}}^k, \widehat{b}^k}^{\pi_{\mathrm{dis}}^*}(s_1, i^* \upsilon) \right)$$

$$= \left( i^* \upsilon - \tau^{-1} \overline{V}_{1, \widehat{\mathcal{P}}^k, \widehat{b}^k}^{\pi_{\mathrm{dis}}^*}(s_1, i^* \upsilon) \right) - \mathsf{CVaR}_\tau(\overline{R}(\pi^k, i^k \upsilon)) + \tau^{-1} \left( \overline{V}_{1, \widehat{\mathcal{P}}^k, \widehat{b}^k}^{\pi_{\mathrm{dis}}^*}(s_1, i^* \upsilon) - \overline{V}_{1, \mathcal{P}^*, 0}^{\pi_{\mathrm{dis}}^*}(s_1, i^* \upsilon) \right) \tag{22}$$

Suppose in $k$-th iteration, *CVaR-LSVI* returns $\widehat{\overline{V}}_{1, \widehat{\mathcal{P}}^k, \widehat{b}^k}^*(s_1, i^* \upsilon)$ for the initial budget $i^* \upsilon$. Then from Theorem B.2, we know

$$\overline{V}_{1, \widehat{\mathcal{P}}^k, \widehat{b}^k}^{\pi_{\mathrm{dis}}^*}(s_1, i^* \upsilon) \geq \widehat{\overline{V}}_{1, \widehat{\mathcal{P}}^k, \widehat{b}^k}^*(s_1, i^* \upsilon) - \frac{\tau}{8} \epsilon.$$

This implies that

$$i^* \upsilon - \tau^{-1} \overline{V}_{1, \widehat{\mathcal{P}}^k, \widehat{b}^k}^{\pi_{\mathrm{dis}}^*}(s_1, i^* \upsilon) \leq i^* \upsilon - \tau^{-1} \widehat{\overline{V}}_{1, \widehat{\mathcal{P}}^k, \widehat{b}^k}^*(s_1, i^* \upsilon) + \frac{\epsilon}{8}$$

$$\leq i^k - \tau^{-1} \widehat{\overline{V}}_{1, \widehat{\mathcal{P}}^k, \widehat{b}^k}^*(s_1, i^k \upsilon) + \frac{\epsilon}{8}$$

$$\leq i^k - \tau^{-1} \overline{V}_{1, \widehat{\mathcal{P}}^k, \widehat{b}^k}^{\pi^k}(s_1, i^k \upsilon) + \frac{\epsilon}{6}, \tag{23}$$

where the last step is due to Theorem 5.1.

On the other hand, from (2) we know

$$\mathsf{CVaR}_\tau(\overline{R}(\pi^k, i^k \upsilon)) \geq i^k - \overline{V}_{1, \mathcal{P}^*, 0}^{\pi^k}(s_1, i^k \upsilon). \tag{24}$$

Plug Equation (23) and (24) into (22), we have

$$\overline{\mathsf{CVaR}}_\tau^* - \mathsf{CVaR}_\tau(\overline{R}(\pi^k, i^k \upsilon)) \leq \tau^{-1} \left( \overline{V}_{1, \mathcal{P}^*, 0}^{\pi^k}(s_1, i^k \upsilon) - \overline{V}_{1, \widehat{\mathcal{P}}^k, \widehat{b}^k}^{\pi^k}(s_1, i^k \upsilon) \right)$$

$$+ \tau^{-1} \left( \overline{V}_{1, \widehat{\mathcal{P}}^k, \widehat{b}^k}^{\pi_{\mathrm{dis}}^*}(s_1, i^* \upsilon) - \overline{V}_{1, \mathcal{P}^*, 0}^{\pi_{\mathrm{dis}}^*}(s_1, i^* \upsilon) \right) + \frac{\epsilon}{6}$$

Then, following the same arguments in the proof of Theorem 4.1, we can obtain

$$\sum_{k=1}^{K} \overline{\mathsf{CVaR}}_\tau^* - \mathsf{CVaR}_\tau(\overline{R}(\pi^k, i^k \upsilon)) \leq \tilde{O}\left( \tau^{-1} H^3 A d^2 \sqrt{K} \cdot \sqrt{\log(|\mathcal{F}|/\delta)} \right) + \frac{K\epsilon}{6}. \tag{25}$$

Next we bridge $\overline{\mathsf{CVaR}}_\tau$ and $\mathsf{CVaR}_\tau$ via Proposition B.1. Note that from Proposition B.1 we have

$$\sum_{k=1}^{K} \mathsf{CVaR}_\tau^* - \mathsf{CVaR}_\tau(R(\overline{\pi}^k, i^k \upsilon)) \leq \sum_{k=1}^{K} \overline{\mathsf{CVaR}}_\tau^* - \mathsf{CVaR}_\tau(\overline{R}(\pi^k, i^k \upsilon)) + \frac{KH\upsilon}{\tau}. \tag{26}$$

Combining (25) and (26), we have

$$\sum_{k=1}^{K} \mathsf{CVaR}_\tau^* - \mathsf{CVaR}_\tau(R(\overline{\pi}^k, i^k \upsilon)) \leq \tilde{O}\left( \tau^{-1} H^3 A d^2 \sqrt{K} \cdot \sqrt{\log(|\mathcal{F}|/\delta)} \right) + \frac{K\epsilon}{6} + \frac{KH\upsilon}{\tau}.$$

Substituting the values of the parameters, we will obtain

$$\frac{1}{K} \sum_{k=1}^{K} \mathsf{CVaR}_\tau^* - \mathsf{CVaR}_\tau(R(\overline{\pi}^k, i^k \upsilon)) \leq \epsilon.$$

This implies that the uniformly mixed policy of $\{(\overline{\pi}^k, i^k \upsilon)\}_{k=1}^{K}$ is $\epsilon$-optimal. The total number of collected trajectories is $KH = \tilde{O}\left( \frac{H^7 A^2 d^4 \log \frac{|\mathcal{F}|}{\delta}}{\tau^2 \epsilon^2} \right)$.

Now, we discuss the computational complexity. The MLE oracle is called $KH$ times. For the rest of the computation, the cost is dominated by running *CVaR-LSVI* in Line 17 of Algorithm 3. In *CVaR-LSVI*, we compute $(\Lambda_h^t)^{-1}$ by the Sherman-Morrison formula, then the computational complexity of *CVaR-LSVI* is dominated by Line 6 of Algorithm 2, which requires $O(\frac{d^2 A T_1}{v^2})$ time per step and leads to a total computational complexity of $O(\frac{d^2 A H^2 T_1^2}{v^3})$ for each call of *CVaR-LSVI*. Note that *CVaR-LSVI* is called totally $\frac{KH}{v}$ times in Algorithm 3 and thus the total computation cost of Algorithm 3 is

$$O\left(\frac{Kd^2 AH^3 T_1^2}{v^4}\right) = \tilde{O}\left(\frac{H^{29} A^3 d^{12} \log \frac{|\mathcal{F}|}{\delta}}{\tau^{16} \epsilon^{16}}\right).$$

### D.3 PROOF OF LEMMA D.1

The proof largely follows the proof of Jin et al. (2020)[Lemma B.3]. We first bound $w_h^t(iv)$. Note that we have for any vector $v \in \mathbb{R}^{\bar{d}}$ where $\bar{d} := d(1 + \lceil 1/v \rceil)$ that

$$\left|v^\top w_h^t(iv)\right| \leq \sum_{j=1}^{t-1} \left|v^\top (\Lambda_h^t)^{-1} \overline{\phi}_h^j\right| \cdot H$$

$$\leq \sqrt{\left(\sum_{j=1}^{t-1} v^\top (\Lambda_h^t)^{-1} v\right) \left(\sum_{j=1}^{t-1} \left(\overline{\phi}_h^j\right)^\top (\Lambda_h^t)^{-1} \overline{\phi}_h^j\right)} \cdot H$$

$$\leq H\|v\| \sqrt{\frac{t}{\lambda}} \cdot \sqrt{\sum_{j=1}^{t-1} \left(\overline{\phi}_h^j\right)^\top (\Lambda_h^t)^{-1} \overline{\phi}_h^j}.$$

Note that with Lemma E.6 we have $\sum_{j=1}^{t-1} \left(\overline{\phi}_h^j\right)^\top (\Lambda_h^t)^{-1} \overline{\phi}_h^j \leq \bar{d}$ and therefore we can obtain that for all $v \in \mathbb{R}^{\bar{d}}$

$$\left|v^\top w_h^t(iv)\right| \leq H\|v\| \sqrt{\frac{t\bar{d}}{\lambda}},$$

which implies that $\|w_h^t(iv)\| \leq H\sqrt{\frac{t\bar{d}}{\lambda}}$.

To utilize Lemma E.7, we further need to bound the covering number of the class of estimated value functions. Fix $h \in [H]$, it can be observed that all $\overline{V}_h^t(\cdot, \cdot)$ where $t \in [T_1]$ belongs to the following function class $\mathcal{V}$ parametrized by $w$ and $A$:

$$\left\{V \mid V(s, iv) = \min_a \text{Clip}_{[-H, H]} \left(\overline{\phi}_h^\top(s, a) w(iv) - \widehat{b}_h(s, a) - \|\overline{\phi}_h(s, a)\|_A\right)\right\},$$

where $\|w(iv)\| \leq H\sqrt{\frac{t\bar{d}}{\lambda}}$ and $\|A\| \leq \beta^2/\lambda$. Then for any $V_1$ (parametrized by $w_1, A_1$) and $V_2$ (parametrized by $w_2, A_2$), we know

$$\sup_{s \in \mathcal{S}, 0 \leq i \leq \lceil H/v \rceil} |V_1(s, iv) - V_2(s, iv)|$$

$$\leq \sup_{s \in \mathcal{S}, a \in \mathcal{A}, 0 \leq i \leq \lceil H/v \rceil} \left|\overline{\phi}_h^\top(s, a)(w_1(iv) - w_2(iv)) - (\|\overline{\phi}_h(s, a)\|_{A_1} - \|\overline{\phi}_h(s, a)\|_{A_2})\right|$$

$$\leq \sup_{\|\overline{\phi}\| \leq 1, 0 \leq i \leq \lceil H/v \rceil} \left|\overline{\phi}^\top (w_1(iv) - w_2(iv))\right| + \sup_{\|\overline{\phi}\| \leq 1} \|\overline{\phi}\|_{A_1 - A_2}$$

$$\leq \sup_{0 \leq i \leq \lceil H/v \rceil} \|w_1(iv) - w_2(iv)\| + \sqrt{\|A_1 - A_2\|_F}.$$

This indicates that the covering number of $\mathcal{V}$ with respect to $\ell_\infty$ norm can be upper bounded by the covering number of $w$ and $A$. More specifically, let $\mathcal{N}(\epsilon)$ denote the covering number of $\mathcal{V}$ with

respect to $\ell_\infty$ norm, then we have

$$\log \mathcal{N}(\epsilon) \leq O\left(\overline{d}(\overline{d} + \frac{H}{\upsilon}) \log \frac{HT_1 \overline{d}}{\epsilon \lambda}\right) \leq O\left(\frac{Hd^2}{\upsilon^2} \log \frac{HT_1 d}{\epsilon \lambda \upsilon}\right).$$

Substituting the above bound into Lemma E.7 concludes the proof.

### D.4 PROOF OF LEMMA D.3

The proof is conducted via induction. For the base case, when $h = H + 1$, we have $\overline{V}_{H+1}^t(s, i\upsilon) = \overline{V}_{H+1}^*(s, i\upsilon) = i\upsilon$ for all $s \in \mathcal{S}$ and $0 \leq i \leq \lceil H/\upsilon \rceil$.

Then suppose we have $\overline{V}_{h+1}^t(s, i\upsilon) \leq \overline{V}_{H+1}^*(s, i\upsilon)$ for all $s \in \mathcal{S}$ and $0 \leq i \leq \lceil H/\upsilon \rceil$. For any $s \in \mathcal{S}, a \in \mathcal{A}$ and $0 \leq i \leq \lceil H/\upsilon \rceil$, if $\overline{Q}_h^t(s, i\upsilon, a) = -H$, then $\overline{Q}_h^t(s, i\upsilon, a) \leq \overline{Q}_h^*(s, i\upsilon, a)$ naturally holds. Otherwise, from Lemma D.2 we know

$$\overline{Q}_h^t(s, i\upsilon, a) - \overline{Q}_h^*(s, i\upsilon, a) \leq -\widehat{b}_h(s, a) + \left(\overline{\phi}_h(s, a)\right)^\top w_h^t(i\upsilon) - \beta \left\|\overline{\phi}_h(s, a)\right\|_{(\Lambda_h^t)^{-1}} - \overline{Q}_h^*(s, i\upsilon, a)$$

$$\leq \left(\widehat{\mathcal{P}}_h \left(\overline{V}_{h+1}^t - \overline{V}_{h+1}^\pi\right)\right)(s, i\upsilon, a) \leq 0.$$

Therefore we have $\overline{Q}_h^t(s, i\upsilon, a) \leq \overline{Q}_h^*(s, i\upsilon, a)$ for all $s \in \mathcal{S}, a \in \mathcal{A}$ and $0 \leq i \leq \lceil H/\upsilon \rceil$, which leads to $\overline{V}_h^t(s, i\upsilon) \leq \overline{V}_h^*(s, i\upsilon)$ for all $s \in \mathcal{S}$ and $0 \leq i \leq \lceil H/\upsilon \rceil$ as well. This concludes our proof.

### D.5 PROOF OF LEMMA D.4

First note that from Lemma D.3 we have $\overline{V}_1^*(s_1, i_1\upsilon) \geq \overline{V}_1^t(s_1, i_1\upsilon)$ for all $t \in [T_1]$. This indicates that conditioned on $\mathcal{E}_1$,

$$\sum_{t=1}^{T_1} \overline{V}_1^{\widetilde{\pi}^t}(s_1, i_1\upsilon) - \overline{V}_1^*(s_1, i_1\upsilon) \leq \sum_{t=1}^{T_1} \overline{V}_1^{\widetilde{\pi}^t}(s_1, i_1\upsilon) - \overline{V}_1^t(s_1, i_1\upsilon). \tag{27}$$

Fix $t \in [T_1]$ and then we know

$$\overline{V}_1^{\widetilde{\pi}^t}(s_1, i_1\upsilon) - \overline{V}_1^t(s_1, i_1\upsilon) = \overline{Q}_1^{\widetilde{\pi}^t}(s_1^t, i_1\upsilon, a_1^t) - \overline{Q}_1^t(s_1, i_1\upsilon, a_1^t).$$

If $\overline{Q}_1^t(s_1^t, i_1\upsilon, a_1^t) = H$, then we know $\overline{Q}_1^{\widetilde{\pi}^t}(s_1^t, i_1\upsilon, a_1^t) - \overline{Q}_1^t(s_1, i_1\upsilon, a_1^t) \leq 0$. Otherwise, we have

$$\overline{Q}_1^{\widetilde{\pi}^t}(s_1^t, i_1\upsilon, a_1^t) - \overline{Q}_1^t(s_1, i_1\upsilon, a_1^t) \leq \overline{Q}_1^{\widetilde{\pi}^t}(s_1^t, i_1\upsilon, a_1^t) + \widehat{b}_h(s_1^t, a_1^t) - \left\langle \overline{\phi}_1^t, w_h^t(i_1\upsilon) \right\rangle + \beta \|\overline{\phi}_1^t\|_{(\Lambda_1^t)^{-1}}.$$

Then with Lemma D.2, we know

$$\overline{Q}_1^{\widetilde{\pi}^t}(s_1^t, i_1\upsilon, a_1^t) - \overline{Q}_1^t(s_1, i_1\upsilon, a_1^t) \leq \left(\widehat{\mathcal{P}}_h \left(\overline{V}_2^{\widetilde{\pi}^t} - \overline{V}_2^t\right)\right)(s_1^t, i_1\upsilon, a_1^t) + 2\beta \|\overline{\phi}_1^t\|_{(\Lambda_1^t)^{-1}}. \tag{28}$$

Let $i_h^t\upsilon$ denote $i_1\upsilon - \sum_{h'=1}^{h-1} \overline{r}_h^t$ and let $\{(\zeta_h^t)_{h=1}^H\}_{t=1}^{T_1}$ denote the following random variable:

$$\zeta_h^t := \begin{cases} 0, & \text{if there exists some } h' \leq h \text{ s.t. } \overline{Q}_{h'}^t(s_{h'}^t, i_{h'}^t\upsilon, a_{h'}^t) = H, \\ \left(\widehat{\mathcal{P}}_h \left(\overline{V}_{h+1}^{\widetilde{\pi}^t} - \overline{V}_{h+1}^t\right)\right)(s_h^t, i_h^t\upsilon, a_h^t) - \left(\overline{V}_{h+1}^{\widetilde{\pi}^t}(s_{h+1}^t, i_{h+1}^t\upsilon) - \overline{V}_2^t(s_{h+1}^t, i_{h+1}^t\upsilon)\right), & \text{otherwise.} \end{cases}$$

It can be easily observed that $\zeta_h^t \in \mathcal{G}_{t,h+1}$ and $\mathbb{E}[\zeta_h^t | \mathcal{G}_{t,h}] = 0$, which means that $\{(\zeta_h^t)_{h=1}^H\}_{t=1}^{T_1}$ is a martingale with respect to $\{\mathcal{G}_{t,h}\}$. Then when $\overline{Q}_1^t(s_1^t, i_1\upsilon, a_1^t) \neq H$, Equation (28) is equivalent to

$$\overline{Q}_1^{\widetilde{\pi}^t}(s_1^t, i_1\upsilon, a_1^t) - \overline{Q}_1^t(s_1, i_1\upsilon, a_1^t) \leq \overline{V}_2^{\widetilde{\pi}^t}(s_2^t, i_2^t\upsilon) - \overline{V}_2^t(s_2^t, i_2^t\upsilon) + \zeta_h^t + 2\beta \|\overline{\phi}_1^t\|_{(\Lambda_1^t)^{-1}}.$$

Repeat such expansion till a step $h_t$ such that $\overline{Q}_{h_t}^t(s_{h_t}^t, i_{h_t}^t\upsilon, a_{h_t}^t) = H$ or $h_t = H + 1$. Then we have

$$\overline{V}_1^{\widetilde{\pi}^t}(s_1, i_1\upsilon) - \overline{V}_1^t(s_1, i_1\upsilon) \leq \sum_{h=1}^H \zeta_h^t + 2\beta \sum_{h=1}^{h_t-1} \|\overline{\phi}_h^t\|_{(\Lambda_h^t)^{-1}} \leq \sum_{h=1}^H \zeta_h^t + 2\beta \sum_{h=1}^H \|\overline{\phi}_h^t\|_{(\Lambda_h^t)^{-1}}. \tag{29}$$

Therefore combining (28) and (29), we know

$$\sum_{t=1}^{T_1} \overline{V}_1^{\widetilde{\pi}^t}(s_1, i_1 v) - \overline{V}_1^*(s_1, i_1 v) \leq \sum_{t=1}^{T_1} \sum_{h=1}^{H} \zeta_h^t + 2\beta \sum_{t=1}^{T_1} \sum_{h=1}^{H} \|\overline{\phi}_h^t\|_{(\Lambda_h^t)^{-1}}. \tag{30}$$

For the first term of the RHS of (30), from Azuma-Hoeffding's inequality, we have with probability at least $1 - \delta/2$ that

$$\sum_{t=1}^{T_1} \sum_{h=1}^{H} \zeta_h^t \leq 2H\sqrt{T_1 H \iota^{\frac{1}{2}}}.$$

For the second term, we can utilize the elliptical potential lemma (Lemma E.8 in Appendix E), which yields

$$\sum_{t=1}^{T_1} \sum_{h=1}^{H} \|\overline{\phi}_h^t\|_{(\Lambda_h^t)^{-1}} \leq \sum_{h=1}^{H} \sqrt{T_1} \cdot \sqrt{\sum_{t=1}^{T_1} \sum_{h=1}^{H} (\overline{\phi}_h^t)^\top (\Lambda_h^t)^{-1} \overline{\phi}_h^t} \leq H\sqrt{\frac{2dT_1 \iota^{\frac{1}{2}}}{v}}.$$

Plug the above two inequalities into (30) leads to the result in Lemma D.4.

# E   AUXILIARY LEMMAS

**Lemma E.1** (MLE guarantees). *For any fixed $(h, k) \in [H] \times [K]$, with probability at least $1 - \delta$, we have that*

$$\mathbb{E}_{s \sim \{0.5\rho_h^k + 0.5\eta_h^k\}, a \sim U(\mathcal{A})}[\|\widehat{P}_h^k(\cdot|s, a) - P_h^*(\cdot|s, a)\|_1^2] \leq \zeta, \ \zeta := \frac{\log(|\mathcal{F}|/\delta)}{k}. \tag{31}$$

*Using union bound, a direct corollary is: with probability at least $1 - \delta$ the following holds for all $h \in [H], k \in [K]$*

$$\mathbb{E}_{s \sim \{0.5\rho_h^k + 0.5\eta_h^k\}, a \sim U(\mathcal{A})}[\|\widehat{P}_h^k(\cdot|s, a) - P_h^*(\cdot|s, a)\|_1^2] \leq 0.5\zeta^k, \ \zeta^k := \frac{\log(|\mathcal{F}|Hk/\delta)}{k}. \tag{32}$$

**Lemma E.2** (Concentration of the bonus term). *Set $\lambda^k = \Theta(d \log(kH|\mathcal{F}|/\delta))$ at the $k$-th episode. Define*

$$\Sigma_{\rho_h^k, \phi} = k\mathbb{E}_{s \sim \rho_h^k, a \sim U(\mathcal{A})}[\phi(s, a)\phi^\top(s, a)] + \lambda^k I, \quad \widehat{\Sigma}_{k, \phi} = \sum_{s, a \in \mathcal{D}_h} \phi(s, a)\phi^\top(s, a) + \lambda^k I.$$

*With probability $1 - \delta$, we have for any $k \in \mathbb{N}^+, h \in [H], \phi \in \Phi$,*

$$\|\phi(s, a)\|_{(\widehat{\Sigma}_{h, \phi}^k)^{-1}} = \Theta\left(\|\phi(s, a)\|_{\Sigma_{\rho_h^k, \phi}^{-1}}\right)$$

**Lemma E.3** (One-step back inequality for the learned model). *Let $\pi = \{\pi_h : \mathcal{S} \times [0, H] \mapsto \Delta(\mathcal{A})\}_{h \in [H]}$ denote any policy on the augmented state. Fix an initial budget $c_1 \in [0, 1]$. Take any $g : \mathcal{S} \times \mathcal{A} \mapsto \mathbb{R}$ such that $\|g\|_\infty \leq B$. We condition on the event where the MLE guarantee (32):*

$$\mathbb{E}_{s \sim \rho_h^k, a \sim U(\mathcal{A})}[f_h^k(s, a)] \lesssim \zeta^k$$

*holds for any $h \in [H]$. For $h = 1$, we have that*

$$\left|\mathbb{E}_{(s, a) \sim d_{1, \widehat{\mathcal{P}}^k}^{(\pi, c_1)}}[g(s, a)]\right| \leq \sqrt{A \cdot \mathbb{E}_{s \sim \rho_1^k, a \sim U(\mathcal{A})}[g^2(s, a)]}$$

*For any $h + 1 \in \{2, \cdots, H\}$ and policy $\pi$, we have that*

$$\left|\mathbb{E}_{(s', a') \sim d_{h+1, \widehat{\mathcal{P}}^k}^{(\pi, c_1)}}[g(s', a')]\right|$$

$$\leq \mathbb{E}_{(s, a) \sim d_{h, \widehat{\mathcal{P}}^k}^{(\pi, c_1)}}\left[\|\widehat{\phi}_h^k(s, a)\|_{\Sigma_{\rho_h^k \times U(\mathcal{A}), \widehat{\phi}_h^k}^{-1}} \sqrt{kA \cdot \mathbb{E}_{s' \sim \eta_{h+1}^k, a' \sim U(\mathcal{A})}[g^2(s', a')] + B^2 \lambda^k d + kB^2 \zeta^k}\right]$$

$$\tag{33}$$

*where $\Sigma_{\rho_h^k \times U(\mathcal{A}), \widehat{\phi}_h^k} = k\mathbb{E}_{s \sim \rho_h^k, a \sim U(\mathcal{A})}[\widehat{\phi}_h^k(\widehat{\phi}_h^k)^\top] + \lambda^k I$.*

*Proof.* For $h = 1$, we have that

$$\mathbb{E}_{(s,a)\sim d_{1,\widehat{\mathcal{P}}^k}^{(\pi,c_1)}}[g(s,a)] \leq \sqrt{\max_{s,a} \frac{d_1(s)\pi_1(a|s,c_1)}{\rho_1^k(s)U(a)}\mathbb{E}_{s\sim\rho_1^k,a\sim U(\mathcal{A})}[g^2(s,a)]}$$

$$\leq \sqrt{A \cdot \mathbb{E}_{s\sim\rho_1^k,a\sim U(\mathcal{A})}[g^2(s,a)]}$$

where we use the fact that $d_1 = \rho_1^k$. Recall that $\rho_h^k(s) = \frac{1}{k}\sum_{i=0}^{k-1} d_{h,c^i}^{\pi^i}(s)$ is the (expected) occupancy of any $s \in \mathcal{S}$ in dataset $\mathcal{D}_h$. Let $\omega_{h,\mathcal{P}}^{(\pi,c_1)} : \mathcal{S} \times \mathcal{A} \mapsto \Delta([0,1])$ denote the distribution of the remaining budget $c_h$ at timestep $h$ conditioned on any $(s,a) \in \mathcal{S} \times \mathcal{A}$ when rolling policy $\pi$ in an MDP with transition kernel $\mathcal{P}$ and the initial budget $c_1$. For any $h \in \{1, \cdots, H-1\}$, we have that

$$\mathbb{E}_{(s',a')\sim d_{h+1,\widehat{\mathcal{P}}^k}^{(\pi,c_1)}}[g(s',a')]$$

$$=\mathbb{E}_{(s,a)\sim d_{h,\widehat{\mathcal{P}}^k}^{(\pi,c_1)}}\mathbb{E}_{s'\sim\widehat{P}_h^k(\cdot|s,a)}\mathbb{E}_{c\sim\omega_{h,\widehat{\mathcal{P}}^k}^{(\pi,c_1)}(\cdot|s,a),r,a'\sim\pi_{h+1}(\cdot|s',c-r)}[g(s',a')]$$

$$=\mathbb{E}_{(s,a)\sim d_{h,\widehat{\mathcal{P}}^k}^{(\pi,c_1)}}\left[\widehat{\phi}_h^k(s,a)^\top \int_{\mathcal{S}} \widehat{\psi}_h^k(s') \cdot \mathbb{E}_{c\sim\omega_{h,\widehat{\mathcal{P}}^k}^{(\pi,c_1)}(\cdot|s,a),r,a'\sim\pi_{h+1}(\cdot|s',c-r)}[g(s',a')]ds'\right]$$

$$\leq\mathbb{E}_{(s,a)\sim d_{h,\widehat{\mathcal{P}}^k}^{(\pi,c_1)}}\left[\|\widehat{\phi}_h^k(s,a)\|_{\Sigma_{\rho_h^k\times U(\mathcal{A}),\widehat{\phi}_h^k}^{-1}}\right.$$

$$\left.\cdot\left\|\int_{\mathcal{S}} \widehat{\psi}_h^k(s') \cdot \mathbb{E}_{c\sim\omega_{h,\widehat{\mathcal{P}}^k}^{(\pi,c_1)}(\cdot|s,a),r,a'\sim\pi_{h+1}(\cdot|s',c-r)}[g(s',a')]ds'\right\|_{\Sigma_{\rho_h^k\times U(\mathcal{A}),\widehat{\phi}_h^k}}\right]$$

where the last inequality is because of CS inequality. For any $(s,a) \in \mathcal{S} \times \mathcal{A}$, we have that

$$\left\|\int_{\mathcal{S}} \widehat{\psi}_h^k(s')\mathbb{E}_{c\sim\omega_{h,\widehat{\mathcal{P}}^k}^{(\pi,c_1)}(\cdot|s,a),r,a'\sim\pi_{h+1}(\cdot|s',c-r)}[g(s',a')]ds'\right\|_{\Sigma_{\rho_h^k\times U(\mathcal{A}),\widehat{\phi}_h^k}}^2$$

$$\leq\left(\int_{\mathcal{S}} \widehat{\psi}_h^k(s')\mathbb{E}_{c\sim\omega_{h,\widehat{\mathcal{P}}^k}^{(\pi,c_1)}(\cdot|s,a),r,a'\sim\pi_{h+1}(\cdot|s',c-r)}[g(s',a')]ds'\right)^\top \cdot \left\{k\mathbb{E}_{\tilde{s}\sim\rho_h^k,\tilde{a}\sim U(\mathcal{A})}[\widehat{\phi}_h^k(\widehat{\phi}_h^k)^\top] + \lambda^k I\right\}$$

$$\cdot\left(\int_{\mathcal{S}} \widehat{\psi}_h^k(s')\mathbb{E}_{c\sim\omega_{h,\widehat{\mathcal{P}}^k}^{(\pi,c_1)}(\cdot|s,a),r,a'\sim\pi_{h+1}(\cdot|s',c-r)}[g(s',a')]ds'\right)$$

$$\leq k \cdot \mathbb{E}_{\tilde{s}\sim\rho_h^k,\tilde{a}\sim U(\mathcal{A})}\left[\left(\int_{\mathcal{S}} \widehat{\psi}_h^k(s')^\top\widehat{\phi}_h^k(\tilde{s},\tilde{a})\mathbb{E}_{c\sim\omega_{h,\widehat{\mathcal{P}}^k}^{(\pi,c_1)}(\cdot|s,a),r,a'\sim\pi_{h+1}(\cdot|s',c-r)}[g(s',a')]ds'\right)^2\right] + B^2\lambda^k d$$

where we use the assumption

$$\|\sum_{a'}\mathbb{E}_{c\sim\omega_{h,\widehat{\mathcal{P}}^k}^{(\pi,c_1)}(\cdot|s,a),r,a'\sim\pi_{h+1}(\cdot|s',c-r)}[g(s',a')]\| \leq B,$$

and

$$\int_{\mathcal{S}}\|\widehat{\psi}_h^k(s')y(s')ds'\|_2 \leq \sqrt{d},$$

for any $y : \mathcal{S} \mapsto [0,1]$. Note that $\widehat{\psi}_h^k(s')\widehat{\phi}_h^k(\tilde{s},\tilde{a}) = \widehat{P}_h^k(s'|\tilde{s},\tilde{a})$. Further, we derive that

$$k \cdot \mathbb{E}_{\tilde{s}\sim\rho_h^k,\tilde{a}\sim U(\mathcal{A})}\left[\left(\int_{\mathcal{S}} \widehat{P}_h^k(s'|\tilde{s},\tilde{a})\mathbb{E}_{c\sim\omega_{h,\widehat{\mathcal{P}}^k}^{(\pi,c_1)}(\cdot|s,a),r,a'\sim\pi_{h+1}(\cdot|s',c-r)}[g(s',a')]ds'\right)^2\right] + B^2\lambda^k d$$

$$\leq k \cdot \mathbb{E}_{\tilde{s}\sim\rho_h^k,\tilde{a}\sim U(\mathcal{A})}\left[\left(\mathbb{E}_{s'\sim P_h^*(\cdot|\tilde{s},\tilde{a})}\mathbb{E}_{c\sim\omega_{h,\widehat{\mathcal{P}}^k}^{(\pi,c_1)}(\cdot|s,a),r,a'\sim\pi_{h+1}(\cdot|s',c-r)}[g(s',a')]\right)^2\right] + B^2\lambda^k d + kB^2\zeta^k$$

$$\leq k \cdot \mathbb{E}_{\tilde{s}\sim\rho_h^k,\tilde{a}\sim U(\mathcal{A}),s'\sim P_h^*(\cdot|\tilde{s},\tilde{a})}\mathbb{E}_{c\sim\omega_{h,\widehat{\mathcal{P}}^k}^{(\pi,c_1)}(\cdot|s,a),r,a'\sim\pi_{h+1}(\cdot|s',c-r)}[g(s',a')]^2 + B^2\lambda^k d + kB^2\zeta^k$$

$$\leq kA \cdot \mathbb{E}_{\tilde{s}\sim\rho_h^k,\tilde{a}\sim U(\mathcal{A}),s'\sim P_h^*(\cdot|\tilde{s},\tilde{a}),a'\sim U(\mathcal{A})}[g^2(s',a')] + B^2\lambda^k d + kB^2\zeta^k$$

Note that $\tilde{s} \sim \rho_h^k, \tilde{a} \sim U(\mathcal{A}), s' \sim P_h^*(\cdot|\tilde{s}, \tilde{a})$ is equivalent to $s' \sim \eta_{h+1}^k$. Therefore,

$$
\mathbb{E}_{(s',a') \sim d_{h+1,\widehat{\mathcal{P}}^k}^{(\pi, c_1)}} [g(s', a')]
$$

$$
\leq \mathbb{E}_{(s,a) \sim d_{h,\widehat{\mathcal{P}}^k}^{(\pi, c_1)}} \left[ \|\widehat{\phi}_h^k(s, a)\|_{\Sigma_{\rho_h^k \times U(\mathcal{A}), \widehat{\phi}_h^k}^{-1}} \left\| \int_{\mathcal{S}} \sum_{a'} \widehat{\psi}_h^k(s') \mathbb{E}_{c, r_h} \left[ \pi_{h+1}(a'|s', c - r_h) \right] g(s', a') ds' \right\|_{\Sigma_{\rho_h^k \times U(\mathcal{A}), \widehat{\phi}_h^k}} \right]
$$

$$
\leq \mathbb{E}_{(s,a) \sim d_{h,\widehat{\mathcal{P}}^k}^{(\pi, c_1)}} \left[ \|\widehat{\phi}_h^k(s, a)\|_{\Sigma_{\rho_h^k \times U(\mathcal{A}), \widehat{\phi}_h^k}^{-1}} \sqrt{kA \cdot \mathbb{E}_{s' \sim \eta_{h+1}^k, a' \sim U(\mathcal{A})}[g^2(s', a')] + B^2 \lambda^k d + kB^2 \zeta^k} \right]
$$

which concludes the proof. □

**Lemma E.4** (One-step back inequality for the true model). *Let $\pi = \{\pi_h : \mathcal{S} \times [0, H] \mapsto \Delta(\mathcal{A})\}_{h \in [H]}$ denote any policy on the augmented state. Fix an initial budget $c_1 \in [0, 1]$. Take any $g : \mathcal{S} \times \mathcal{A} \mapsto \mathbb{R}$ such that $\|g\|_\infty \leq B$. Then for $h = 1$, we have that*

$$
\mathbb{E}_{(s,a) \sim d_1^{(\pi, c_1)}}[g(s, a)] \leq \sqrt{A \cdot \mathbb{E}_{s \sim \rho_1^k, a \sim U(\mathcal{A})}[g^2(s, a)]}
$$

*For $h + 1 \in \{2, \cdots, H\}$, we have that*

$$
\mathbb{E}_{(s',a') \sim d_{h+1}^{(\pi, c_1)}}[g(s', a')] \leq \mathbb{E}_{(s,a) \sim d_h^{(\pi, c_1)}} \left[ \|\phi_h^*(s, a)\|_{\Sigma_{\rho_h^k, \phi^*}^{-1}} \sqrt{kA \cdot \mathbb{E}_{s' \sim \rho_{h+1}^k, a' \sim U(\mathcal{A})}[g^2(s', a')] + \lambda^k dB^2} \right]
$$

*Proof.* For $h = 1$, we have that

$$
\mathbb{E}_{(s,a) \sim d_1^{(\pi, c_1)}}[g(s, a)] \leq \sqrt{\frac{d_1(s) \pi_1(a|s, c_1)}{\rho_1^k(s) U(a)} \mathbb{E}_{s \sim \rho_1^k, a \sim U(\mathcal{A})}[g^2(s, a)]} \leq \sqrt{A \cdot \mathbb{E}_{s \sim \rho_1^k, a \sim U(\mathcal{A})}[g^2(s, a)]}
$$

Let $\omega_h^{(\pi, c_1)} := \omega_{h, \mathcal{P}^*}^{(\pi, c_1)}$ denote the distribution of the remaining budget $c_h$ at timestep $h$ conditioned on any $(s, a) \in \mathcal{S} \times \mathcal{A}$ when rolling policy $\pi$ in the (true) environment and the initial budget $c_1$. For $h + 1 \in \{2, \cdots, H\}$, by CS inequality, we derive that

$$
\mathbb{E}_{(s',a') \sim d_{h+1}^{(\pi, c_1)}}[g(s', a')]
$$

$$
= \mathbb{E}_{(s,a) \sim d_h^{(\pi, c_1)}, s' \sim P_h^*(\cdot|s,a)} \mathbb{E}_{c \sim \omega_h^{(\pi, c_1)}, r, a' \sim \pi_{h+1}(\cdot|s', c-r)}[g(s', a')]
$$

$$
\leq \mathbb{E}_{(s,a) \sim d_h^{(\pi, c_1)}} \left[ \|\phi_h^*(s, a)\|_{\Sigma_{\rho_h^k, \phi^*}^{-1}} \left\| \int_{\mathcal{S}} \psi_h^*(s') \mathbb{E}_{c \sim \omega_h^{(\pi, c_1)}, r, a' \sim \pi_{h+1}(\cdot|s', c-r)}[g(s', a')] ds' \right\|_{\Sigma_{\rho_h^k, \phi^*}} \right]
$$

For any $(s, a) \in \mathcal{S} \times \mathcal{A}$, we obtain that

$$
\left\| \int_{\mathcal{S}} \psi_h^*(s') \mathbb{E}_{c \sim \omega_h^{(\pi, c_1)}, r, a' \sim \pi_{h+1}(\cdot|s', c-r)}[g(s', a')] ds' \right\|_{\Sigma_{\rho_h^k, \phi^*}}^2
$$

$$
\leq \left\{ \int_{\mathcal{S}} \psi_h^*(s') \mathbb{E}_{c \sim \omega_h^{(\pi, c_1)}, r, a' \sim \pi_{h+1}(\cdot|s', c-r)}[g(s', a')] ds' \right\}^\top \left\{ k \mathbb{E}_{(\tilde{s}, \tilde{a}) \sim \rho_h^k} [\phi_h^*(\phi_h^*)^\top] + \lambda^k I \right\}
$$

$$
\cdot \left\{ \int_{\mathcal{S}} \psi_h^*(s') \mathbb{E}_{c \sim \omega_h^{(\pi, c_1)}, r, a' \sim \pi_{h+1}(\cdot|s', c-r)}[g(s', a')] ds' \right\}
$$

$$
\leq k \mathbb{E}_{(\tilde{s}, \tilde{a}) \sim \rho_h^k, s' \sim P_h^*(\cdot|\tilde{s}, \tilde{a})} \mathbb{E}_{c \sim \omega_h^\pi, r, a' \sim \pi_{h+1}(a', s', c-r)}[g^2(s', a')] + \lambda^k dB^2
$$

$$
\leq kA \cdot \mathbb{E}_{s' \sim \rho_{h+1}^k, a' \sim U(\mathcal{A})}[g^2(s', a')] + \lambda^k dB^2
$$

where the last inequality holds by the fact that $(\tilde{s}, \tilde{a}) \sim \rho_h^k, s' \sim P_h^*(\tilde{s}, \tilde{a})$ is equivalent to $s' \sim \rho_{h+1}^k$ and importance sampling. Therefore, we conclude the proof. □

**Lemma E.5** (Simulation lemma for risk-neutral RL). *Let $\widehat{\mathcal{M}}$ and $\mathcal{M}$ be to MDPs with the same state and action spaces, but they have different reward and transition functions, i.e., $(\widehat{r}_h, \widehat{P}_h)_{h=1}^H$ and*

$(r_h, P_h)_{h=1}^{H}$, respectively. Consider any policy $\pi = \{\pi_h : \mathcal{S} \to \Delta(\mathcal{A})\}_{h \in [H]}$, denote $\{V_{h,\widehat{\mathcal{M}}}^{\pi}\}_{h=1}^{H}$ and $\{V_h^{\pi}\}_{h=1}^{H}$ be the value functions. Then we have that

$$V_{1,\widehat{\mathcal{M}}}^{\pi}(s_1) - V_1^{\pi}(s_1) = \sum_{h=1}^{H} \mathbb{E}_{(s,a) \sim d_{h,\widehat{\mathcal{M}}}^{\pi}} \left( \widehat{r}_h(s,a) - r_h(s,a) + \left\langle \widehat{P}_h(\cdot|s,a) - P_h(\cdot|s,a), V_{h+1}^{\pi}(\cdot) \right\rangle \right)$$

*Equivalently, we have that*

$$V_{1,\widehat{\mathcal{M}}}^{\pi}(s_1) - V_1^{\pi}(s_1) = \sum_{h=1}^{H} \mathbb{E}_{(s,a) \sim d_h^{\pi}} \left( \widehat{r}_h(s,a) - r_h(s,a) + \left\langle \widehat{P}_h(\cdot|s,a) - P_h(\cdot|s,a), V_{h+1,\widehat{\mathcal{M}}}^{\pi}(\cdot) \right\rangle \right)$$

*where $V_{H+1}^{\pi} = \widehat{V}_{H+1,\widehat{\mathcal{M}}}^{\pi} = 0$ (as the episode ends at $H$).*

*Proof.* Recall that $d_h^{\pi}(s,a)$ and $d_{h,\widehat{\mathcal{M}}}^{\pi}(s,a)$ are the occupancy measures of any $(s,a) \in \mathcal{S} \times \mathcal{A}$ at timestep $h \in [H]$ when executing policy $\pi$ in $\mathcal{M}$ and $\widehat{\mathcal{M}}$, respectively. By definition, we have that

$$V_{1,\widehat{\mathcal{M}}}^{\pi}(s_1) - V_1^{\pi}(s_1)$$

$$= \sum_a \pi_1(s_1,a) \left( \widehat{r}_1(s_1,a) - r_1(s_1,a) + \sum_{s'} \left( \widehat{P}_1(s'|s_1,a)\widehat{V}_{2,\widehat{\mathcal{M}}}^{\pi}(s') - P_1(s'|s_1,a)V_2^{\pi}(s') \right) \right)$$

$$= \mathbb{E}_{(s,a) \sim d_{1,\widehat{\mathcal{M}}}^{\pi}} \left[ \widehat{r}_1(s,a) - r_1(s,a) + \left\langle \widehat{P}_1(\cdot|s,a), \widehat{V}_{2,\widehat{\mathcal{M}}}^{\pi}(\cdot) - V_2^{\pi}(\cdot) \right\rangle + \left\langle \widehat{P}_1(\cdot|s,a) - P_1(\cdot|s,a), V_2^{\pi}(\cdot) \right\rangle \right]$$

$$= \cdots = \sum_{h=1}^{H} \mathbb{E}_{(s,a) \sim d_{h,\widehat{\mathcal{M}}}^{\pi}} \left[ \widehat{r}_h(s,a) - r_h(s,a) + \left\langle \widehat{P}_h(\cdot|s,a) - P_h(\cdot|s,a), V_{h+1}^{\pi}(\cdot) \right\rangle \right]$$

Equivalently, we have that

$$V_{1,\widehat{\mathcal{M}}}^{\pi}(s_1) - V_1^{\pi}(s_1)$$

$$= \mathbb{E}_{(s,a) \sim d_1^{\pi}} \left[ \widehat{r}_1(s,a) - r_1(s,a) + \left\langle P_1(\cdot|s,a), \widehat{V}_{2,\widehat{\mathcal{M}}}^{\pi}(\cdot) - V_2^{\pi}(\cdot) \right\rangle + \left\langle \widehat{P}_1(\cdot|s,a) - P_1(\cdot|s,a), \widehat{V}_{2,\widehat{\mathcal{M}}}^{\pi}(\cdot) \right\rangle \right]$$

$$= \cdots = \sum_{h=1}^{H} \mathbb{E}_{(s,a) \sim d_h^{\pi}} \left[ \widehat{r}_h(s,a) - r_h(s,a) + \left\langle \widehat{P}_h(\cdot|s,a) - P_h(\cdot|s,a), \widehat{V}_{h+1,\widehat{\mathcal{M}}}^{\pi}(\cdot) \right\rangle \right]$$

which concludes the proof. $\square$

**Lemma E.6.** *Let $\Lambda_t = \lambda I + \sum_{j=1}^{t} \overline{\phi}^j (\overline{\phi}^j)^{\top}$ where $\overline{\phi}_i \in \mathbb{R}^d, \lambda > 0, d > 0$, then we have*

$$\sum_{j=1}^{t} (\overline{\phi}^j)^{\top} \Lambda_t^{-1} \overline{\phi}^j \le d.$$

*Proof.* Please refer to (Jin et al., 2020, Lemma D.1). $\square$

**Lemma E.7.** *Let $\{x_j\}_{j=1}^{\infty}$ be a stochastic process on the space $(\mathcal{S}, \{iv\}_{i=0}^{\lceil H/\upsilon \rceil})$ with corresponding filtration $\{\mathcal{G}_j\}_{j=0}^{\infty}$. Let $\{\overline{\phi}_j\}_{j=0}^{\infty}$ be an $\mathbb{R}^{\overline{d}}$-valued stochastic process where $\overline{\phi}_j \in \mathcal{G}_{j-1}$, and $\|\overline{\phi}_j\| \le 1$. Let $\Lambda_t = \lambda I + \sum_{j=1}^{t-1} \overline{\phi}_j \overline{\phi}_j^{\top}$. Then for any $\delta > 0$, with probability at least $1 - \delta$, for all $t \ge 0$, and any $V \in \mathcal{V}$ so that $\sup_x |V(x)| \le H$, we have:*

$$\left\| \sum_{j=1}^{t-1} \overline{\phi}_j \{ V(x_j) - \mathbb{E}[V(x_j)|\mathcal{G}_{j-1}] \} \right\|_{\Lambda_t^{-1}}^2 \le 4H^2 \left[ \frac{\overline{d}}{2} \log \left( \frac{t + \lambda}{\lambda} \right) + \log \frac{\mathcal{N}_{\epsilon}}{\delta} \right] + \frac{8t^2 \epsilon^2}{\lambda},$$

*where $\mathcal{N}_{\epsilon}$ is the $\epsilon$-covering number of $\mathcal{V}$ with respect to the $\ell_{\infty}$-norm $\sup_x |V(x) - V'(x)|$.*

*Proof.* Please refer to (Jin et al., 2020, Lemma D.4). $\square$

**Lemma E.8.** *Suppose $\overline{\phi}_t \in \mathbb{R}^{\overline{d}}$ and $\|\overline{\phi}_t\| \leq 1$ for all $t \geq 0$. For any $t \geq 0$, we define $\Lambda_t = I + \sum_{j=1}^{t} \overline{\phi}_j^\top \overline{\phi}_j$. Then we have*

$$\sum_{j=1}^{t} \overline{\phi}_j^\top \Lambda_{j-1}^{-1} \overline{\phi}_j \leq 2 \log \left[ \frac{\det(\Lambda_t)}{\det(\Lambda_0)} \right].$$

*Proof.* Please refer to (Jin et al., 2020, Lemma D.2). □

