# OpenReview forum: "Provably Efficient CVaR RL in Low-rank MDPs"
_ICLR.cc/2024/Conference — ICLR 2024 poster_

### Official Review · Reviewer_hVSJ · 2023-10-29

**Soundness:** 3 good
**Presentation:** 3 good
**Contribution:** 2 fair
**Rating:** 6
**Confidence:** 4

**Summary:**

This paper study risk-sensitive Reinforcement Learning under low-rank MDPs, where the transitions of MDPs admit a low-rank decomposition into two unknown low-dimension embedding functions and the goal is to maximize the conditional value at risk (CVaR) with certain risk $\tau$. This paper first propose an efficient upper confidence bound type algorithm and then provide regret bound $\tilde{\mathcal{O}}(\frac{H^7A^2d^4}{\tau^2\epsilon^2})$. In addition, this paper disigns a computational efficient LSVI algorithm for planning.

**Strengths:**

1. This paper propose an agorithm with theoretical regret bound, which is new for CVaR RL under low-rank MDPs.

2. This paper propose an computational efficient algorithm for planning.

3. The presentation of this paper is easy to follow and clear notations tables are provided.

**Weaknesses:**

1. My main concern is that the algorithm 1 is quite close to [1]. It seems that algorithm 1 is just an application of REP-UCB in Augmented MDPs. The analysis of regret bound is also similar to [1], with only difference to convert CVaR into value function (as Eq. (13) in Page 17), which makes the result not so surprising and significant. It is likely that I miss some novel analysis techniques, so please bring it out and I think it is also important to stress the novelty in the paper.

2. The authors claim the worse dependency of regret bound on $H^7$ is due to the non-stationary transitions in Page 7. I want to point out it may be wrong. Another previous work considers regular low-rank MDPs with finite horizons and time-dependent transition kernels [2], which is the same as this paper, even has better dependence on $H^3$. It is likely that the worse dependency on $H$ of this paper is due that [1,2] assume that the accumulative discounted reward is in $[0,1]$ and this paper only assumes $[0,H]$. I guess that applying similar assumptions and techniques to clip value functions in [2], the regret bound may acheive the same dependence on $H^3$.

3. Based on the two points above, I think the authors can try to enhence this paper by buliding a lower bound of CVaR RL under low-rank MDPs with better dependence on $\tau$, or try to improve the algorithm to better regret bound.

[1] Masatoshi Uehara, Xuezhou Zhang, and Wen Sun. Representation learning for online and offline RL in low-rank MDPs. In International Conference on Learning Representations, 2022

[2] Cheng, Y., Huang, R., Liang, Y., & Yang, J. Improved Sample Complexity for Reward-free Reinforcement Learning under Low-rank MDPs. In The Eleventh International Conference on Learning Representations, 2023.

**Questions:**

Could the authors also provide some motivation examples of CVaR RL under low-rank MDPs? It will help the readers to understand the significance of this setting and how this formulation is related to real-world applications.

---

> ### Author Response · Authors · 2023-11-17
> **Response to Reviewer hVSJ**
>
> We thank you for taking the time to read our paper and giving valuable comments. Below, we address your concerns about our work.
>
> **novel analysis techniques**: Thanks for the inquiry. Indeed, we acknowledge that our research is built on the foundational work of Wang et al., 2023, and Uehara et al., 2021. While we have taken insights from these significant studies, our work also contributes new techniques that we feel are substantial advancements in the field.
>
> * Firstly, our algorithm achieves efficient exploration while performing representation learning, therefore answering the open problem raised by Wang et al., 2023. Technically, such a tradeoff is hard because we need to carefully maintain two time-dependent datasets (because we study the finite-horizon episodic setting) that provide a diversified exploration. Please see more details about the sampling in our Response to Reviewer htXD.
> * Secondly, our discretized planning oracle, CVaR-LSVI, requires new techniques.
> First, we need to show that LSVI works for CVaR with continuous reward. This is not trivial because the augmented MDP involves the evolution of the budget, which can be non-linear.  Second, we also need to establish a new concentration inequality for CVaR-LSVI (lemma D.1) because we introduce the new budget variable in the regression and need to compute its covering number.
> We believe this oracle's design can benefit the whole risk-averse RL community.
>
> Considering the points mentioned, we maintain that our contributions are meaningful and substantially enrich the current research landscape. In the updated manuscript, we will more strongly emphasize these contributions. Please also refer to our response to Reviewer i64s for more details. We are more than happy to clarify any concerns.
>
> **Lower bound**: Thanks for raising this. We agree that showing lower bounds could be a very interesting direction. Please refer to our response to Reviewer i64s for more details.
>
> **Dependency on $H^7$**: We agree with the reviewer that if we use the $[0,1]$ reward assumption and the corresponding clipping, our bound can be improved to $H^5$. Besides, We think it might be possible to reduce another $H^2$ by utilizing a Bernstein-type bonus in the algorithm instead of the Hoeffding-type bonus, which we leave for future work. Thanks for raising this! We will add clarifications and discussions on the referred papers in the revised version.
>
> **Motivation examples**: Thanks for pointing this out. We totally agree that introducing more real-world motivating examples can significantly benefit our presentation. In fact, the family of low-rank MDPs includes models such as linear MDPs [1] and block MDPs [2], which are widely used in practice. For example, block MDPs are considered in [3, 4] to tackle rich and noisy observations in the environment. In addition, the CVaR metric is popular in robust RL [5], distributional RL [6], and portfolio optimization [7]. Due to their abundant applications, we believe studying the sample complexity of CVaR-RL is of unique interest. We will add more discussions in the next version.
>
> [1] Jin, Chi, et al. "Provably efficient reinforcement learning with linear function approximation." Conference on Learning Theory. PMLR, 2020.
>
> [2] Du, Simon, et al. "Provably efficient RL with rich observations via latent state decoding." International Conference on Machine Learning. PMLR, 2019.
>
> [3] Han, Beining, et al. "Learning domain invariant representations in goal-conditioned block mdps." Advances in Neural Information Processing Systems 34 (2021): 764-776.
>
> [4] Efroni, Yonathan, et al. "Provable RL with exogenous distractors via multistep inverse dynamics." arXiv preprint arXiv:2110.08847 (2021).
>
> [5] Hiraoka, Takuya, et al. "Learning robust options by conditional value at risk optimization." Advances in Neural Information Processing Systems 32 (2019).
>
> [6] Achab, Mastane, et al. "Distributional deep Q-learning with CVaR regression." Deep Reinforcement Learning Workshop NeurIPS 2022. 2022.
>
> [7] Krokhmal, Pavlo, Jonas Palmquist, and Stanislav Uryasev. "Portfolio optimization with conditional value-at-risk objective and constraints." Journal of risk 4 (2002): 43-68.

---

> > ### Comment · Reviewer_hVSJ · 2023-11-18
> >
> > Thanks for the response. Your answer address most of my concerns and I have raised my score.
> >
> > However, I would like to highlight two remaining questions.
> >
> > First, for the novelty about efficient exploration, what I want to understand is that whether, and in what ways, the CVaR will influence efficient exploration under low-rank MDPs, and the difficulty CVaR may bring. Specifically, I want to see some explanation like your second point about "discretized planning oracle". However, I do not find any difference based on your current explanation. Previous research has demonstrated efficient exploration and the utilization of two time-dependent datasets under low-rank MDPs (see my reference above [1,2]).
> >
> > Second, I do not understand "reduce another $H^2 by utilizing a Bernstein-type bonus in the algorithm instead of the Hoeffding-type bonus". In my opinion, the bonus term of low-rank MDPs is not generated by hoeffding type inequality.

---

> ### Author Response · Authors · 2023-11-21
>
> Thank you for raising the score and for bringing up these additional questions.  Please find our responses below.
>
> **Difficulty of CVaR in low-rank MDPs**: We note that the primary challenge in addressing efficient exploration in CVaR-RL is to manage the *continuous* and *history-dependent* nature of the risk budget, which significantly deviates from the risk-neutral setting considered in the previous work and calls for novel techniques. The previous response explains the continuous nature and corresponding solution (discretized LSVI oracle). Here, we elaborate on the challenge that history-dependent nature brings.
>
> * The augmented policy and budget are history-dependent in the original MDPs, making it a complex task to bound the expectation of the estimation error of the transition kernel at timestep $h+1$ under the augmented policy, based only on statistics from timestep $h$ (Lemmas E.3 and E.4). This challenge is distinct from the more straightforward scenario in a risk-neutral setting, where the policy is Markovian and such a bounding process is more apparent.
> * More specifically,  In standard MDPs, they only consider Markovian policy, so when they prove Lemma E.3 and E.4, the next action $a'$ will only depend on the current state $s'$, and they utilize this property to decouple the feature vector $\phi(s, a)$ and function $g(s', a')$.
> * However, in CVaR, we consider augmented policy; thus, the next action $a'$ will also depend on the reward $r$ and budget $c$. Note that both $r$ and $c$ depend on the last state $s$ and action $a$. Consequently, one can not process the decoupling in augmented MDPs. This difference makes our analysis more challenging and non-trivial.
> * To bypass the difficulty, we introduce the quantity $\omega_{h, \mathcal{P}}^{(\pi, c_1)}$ to keep track of the remaining budget, conditioned on which action $a_{h+1}$ is independent of the history, and build our new proof based on this new quantity.  This novel and essential modification helps us overcome the history-dependent challenge.
>
>
> **Hoeffding-type bonus**: Thanks for raising this! About this claim, please find more details below.
>
> * We follow the naming in [3]. Intuitively, we note that our bonus resembles the standard Hoeffding-type bonus term $\widetilde{O}(\sqrt{\frac{1}{N_h(s_h,a_h)}})$ in tabular MDPs [1], where $N_h(s, a)$ is the visitation to $(h,s, a)$. Specifically, the embedding functions in tabular MDPS are given by $\phi_h(s_h,a_h)=\mathbb{1}(s_h,a_h)\in\mathbb{R}^{SA}$ and $\psi_h(s_{h+1})=[P_h^*(s_{h+1}|s_h,a_h)]_{(s_h,a_h)} \in \mathbb{R}^{SA}$.
>
>    Hence, the covariance matrix $\widehat{\Sigma}\_h$ is a $SA$ by $SA$ diagonal matrix, where the diagonal elements are the
>   visitation $N_h$ (when ignoring the regularization term $\lambda^kI_{SA}$). Consequently, the bonus term translates to
>   $\widetilde{O}(\sqrt{\frac{1}{N_h(s_h,a_h)}})$.
>
> * Furthermore, as recent works [2, 3] show that the Bernstein-type bonus works better than the Hoeffding-type bonus in linear MDPs, we anticipate such improvements are also possible for low-rank MDPs. However, directly adopting the techniques may not work since low-rank MDPs are inherently different in the need for learning the embedding functions. Still, it is an interesting topic for future extensions of our work.
>
> Thanks for raising valuable questions. We will add discussions on both points in the revised version.
>
>
> [1] Jin, Chi, et al. "Is Q-learning provably efficient?." Advances in neural information processing systems 31 (2018).
>
> [2] Hu, Pihe, Yu Chen, and Longbo Huang. "Nearly minimax optimal reinforcement learning with linear function approximation." International Conference on Machine Learning. PMLR, 2022.
>
> [3] Wu, Yue, Dongruo Zhou, and Quanquan Gu. "Nearly minimax optimal regret for learning infinite-horizon average-reward MDPs with linear function approximation." International Conference on Artificial Intelligence and Statistics. PMLR, 2022.

---

### Official Review · Reviewer_yZ3N · 2023-10-31

**Soundness:** 3 good
**Presentation:** 3 good
**Contribution:** 2 fair
**Rating:** 6
**Confidence:** 3

**Summary:**

This paper studies risk-sensitive Reinforcement Learning (RL) in the context of low-rank Markov Decision Processes (MDPs), employing the Conditional Value at Risk (CVaR) risk metric. The proposed algorithm bears resemblance to value iteration techniques, incorporating exploration bonuses based on the Upper Confidence Bound (UCB) principle, a well-established concept within the field of RL. Furthermore, the authors introduce a discretized variant of the least-squares value iteration approach and demonstrate its capacity to approach near-optimal policy solutions within a polynomial computational time framework, given access to a Maximum Likelihood Estimation (MLE) oracle. This novel contribution enhances the body of knowledge in this area of research, delivering valuable insights to the academic community.

**Strengths:**

(+) The paper is generally well-written and the technical exposition is overall sound.

(+) I personally find the computationally efficient algorithm a nice addition to the paper. I do not see any citations in Section 5 so I assume that these are novel contributions of the authors.

**Weaknesses:**

(-) In Section 4.1, it is mentioned that the algorithm requires two sets of trajectories. Is it true that a *simulator is required* to obtain these two sets? By simulator, I meant the ability to draw samples from the *true* transition probability $P^*_h$. If this is indeed required, it is a major assumption that the authors fail to disclose in the abstract or in a formal statement. I strongly encourage the authors to highlight this fact.

(-) Again, suppose my understanding of the above is correct. In that case, the result becomes fairly trivial: combining the realizability assumptions with a finite model class (assumption 3.2) with access to a simulator, it is easy to see that given enough samples, any algorithm is no-regret. Nevertheless, there is still credit in proving finite-sample regret bounds in Theorem 4.1.

(-) The claim that this is the first provably efficient CVaR RL algorithm is incorrect. There have been regret bounds for CVaR RL in function approximation settings in the literature.

(-) I also encourage the author to at least discuss the difference between dynamic and static CVaR, as this has non-trivial implications in algorithm design and theoretical analysis.

**Questions:**

Please see the above section and correct me if I misunderstood some results. I am happy to increase my score if that is the case.

---

> ### Author Response · Authors · 2023-11-17
> **Response to Reviewer yZ3N**
>
> We thank you for appreciating our paper presentation and contributions. Below, we address your comments about our work.
>
> **computationally efficient algorithm**: Thanks for appreciating our results! Yes, our results in Section 5.1 provide the first computationally efficient oracle for CVaR RL in the function approximation setting. As mentioned (please refer to our response to Reviewer i64s), strategic planning with risk measures is challenging because every policy is associated with an initial budget in augmented MDP. Therefore, a naive min-max optimization would bring significant computational overheads. Therefore, our discretized LSVI can be widely adopted by the risk-averse RL community.
>
> **Simulator required**: We want to note that our setting is different from the literature of RL with simulators. Usually, in RL problems, the assumption of the simulator indicates that one can input **any** state-action pair into the simulator, and then the next state would be returned.
>
> Instead, our setting is aligned with the online RL literature, which only requires **interaction** with the environment (i.e., one can roll out a policy in the environment) and doesn’t need the simulator.
>
> **any algorithm is no-regret**: We note that the reviewer may have misunderstood our setting. As we consider the classical online RL, the learner has to interact with the environment to optimize the policy. We do not have a simulator that can return the next state for arbitrary state-action input pairs.
>
> It is highly nontrivial to improve the sample complexity in online RL, as shown in the literature (UCB-VI [2], linear MDP [3], bilinear class [4]).  In this case, the notion of no-regret learning refers to attaining a sublinear regret, which can not achieved by an arbitrary algorithm. For example, it is well-known that myopic exploration (e.g., $\epsilon$-greedy) has exponential sample complexity in the worst case [1].
>
> [1] Osband, I., Van Roy, B., Russo, D. J., Wen, Z., et al. Deep exploration via randomized value functions. Journal of Machine Learning Research, 20(124):1–62, 2019.
>
> [2] Azar, M. G., Osband, I., & Munos, R. Minimax regret bounds for reinforcement learning. In International Conference on Machine Learning (pp. 263-272). PMLR.
>
> [3] Jin, Chi, et al. "Provably efficient reinforcement learning with linear function approximation." Conference on Learning Theory. PMLR, 2020.
>
> [4] Du, S., Kakade, S., Lee, J., Lovett, S., Mahajan, G., Sun, W., & Wang, R. Bilinear classes: A structural framework for provable generalization in RL. ICML 2021
>
> **The first regret bounds for CVaR RL**: To the best of our knowledge, we present the first sample-efficient algorithm for optimizing the (static) CVaR metric that carefully balances the interplay between risk-averse RL and low-rank MDP structure. Furthermore, we design a computationally efficient planning oracle that makes our algorithm only require polynomial running time with an MLE oracle.
> * We are aware that [7] provides regret guarantees for dynamic CVaR with general function approximation. However, dynamic CVaR is very different from static CVaR. As you have mentioned, these two metrics have non-trivial implications in algorithm design and theoretical analysis.
> * We would greatly appreciate it if more related works presenting regret guarantees for static CVaR RL with function approximation could be provided. We are more than happy to soften our contribution arguments. Thanks for raising this.
>
> **Difference between dynamic and static CVaR**: The static CVaR setting we studied is quite different from dynamic CVaR, so comparisons are not covered. Thanks for raising this! We plan to add the following paragraph discussing dynamic CVaR.
>
> * Iterated CVaR, also known as dynamic CVaR, is an important variant of CVaR quantifying the worst $\tau$-percent performance \emph{at each step} of decision-making [5-7]. Such a definition allows the agent to control the risk throughout the decision process tightly. Work [6] studied regret minimization of iterated CVaR in tabular MDPs. A recent attempt [7] studies Iterated CVaR RL with general function approximations and provides regret guarantees using the *eluder dimension* of the function class.
>
> It would be greatly appreciated if you could provide additional valuable references. We are happy to add discussions to our work for a more comprehensive review in the next version.
>
>
> [5] Chu, S., & Zhang, Y. (2014). Markov decision processes with iterated coherent risk measures. International Journal of Control, 87(11), 2286-2293.
>
> [6] Du, Y., Wang, S., & Huang, L. Provably Efficient Risk-Sensitive Reinforcement Learning: Iterated CVaR and Worst Path. ICLR 2022
>
> [7] Chen, Y., Du, Y., Hu, P., Wang, S., Wu, D., & Huang, L. (2023). Provably Efficient Iterated CVaR Reinforcement Learning with Function Approximation. arXiv preprint arXiv:2307.02842.

---

> > ### Comment · Reviewer_yZ3N · 2023-11-22
> > **Thank you**
> >
> > Dear authors,
> >
> > I am still confused: without a simulator, at time $h$, how do we take two actions simultaneously? Take playing chess as an example, how do you make two moves at the same time to observe the opponent's reactions to your two moves?
> >
> > Warmest regards,
> > Reviewer.

---

> > > ### Author Response · Authors · 2023-11-22
> > >
> > > Thanks for your follow-up question. We note that our algorithm takes two *consecutive* actions $a_h$ and $\widetilde{a}_{h+1}$ at timesteps $h$ and $h+1$, but *not* taking two actions simultaneously at timestep $h$. Indeed, from Line 8 of Algorithm 1, ELA adds tuple $(s_h,a_h,\tilde{s}\_{h+1})$ to dataset $\mathcal{D}\_h$ and adds tuple $(\tilde{s}\_{h+1},\tilde{a}\_{h+1},s'\_{h+2})$ to dataset $\widetilde{D}\_{h+1}$. Here, the first tuple is collected by taking action $a_h$ at timestep $h$ and the second tuple is collected by taking action $\tilde{a}\_{h+1}$ at timestep $h+1$. Hence, these two actions $a_h$ and $\tilde{a}\_{h+1}$ are taken consecutively, but not simultaneously.

---

### Official Review · Reviewer_i64s · 2023-10-31

**Soundness:** 3 good
**Presentation:** 3 good
**Contribution:** 2 fair
**Rating:** 6
**Confidence:** 3

**Summary:**

In this work, the authors propose the Representation Learning for CVaR (ELA) algorithm for maximizing the Conditional Value at Risk (CVaR) in low rank Markov decision processes (MDPs). The authors then propose a modification of the ELA algorithm, called ELLA, that improves upon the computational complexity of the former. The authors then provide probably approximate correct (PAC) guarantees for both ELA and ELLA.

**Strengths:**

The authors do a good job in introducing both risk sensitive RL and the CVaR objective in Low Rank MDPs. The main result for the ELA algorithm and its ensuing discussion are also well-written. Furthermore the authors partially address one the problems posed in Wang et al, 2023 [1] on whether their minimax CVaR guarantees for tabular MDPs can be extended to low rank MDPs.

Also the introduction and related works do a nice job of setting up the problem and informing the reader on the current state of CVaR and risk-sensitive RL.

**Weaknesses:**

The main weakness of this work lies in the tightness of their guarantees. While the authors provide upper bounds for their ELA and ELLA algorithms, they do not provide lower bounds for CVaR in low rank MDPs. Therefore it is hard to see whether the bound the authors propose is reasonable (see questions for further details).

Another weakness is in the writing of section 4.1. This is a very weird policy to me. First the agent follows the policy to state $s_h$ then takes a uniform action $a_h \sim U(\mathcal{A})$ and receives the next state $s_{h-1} \sim P_h(\cdot | s_h,a_h)$. If the agent is taking uniform actions in step $h$ then how is it following policy $\pi_k$. Wasn't $s_h$ the state the resulted in taking a uniform action $a_{h-1}$ in the previous state $s_h$? Anyways maybe I missed something here but this seems very weird?

**Questions:**

Aside from the questions mentioned in the weaknesses section, I also want to ask where the novelty in this work lies. This work seems like it combines the setting and results of Wang et al, 2023 [1] and Uehara et al, 2021 [2]. What, if anything, needs to change? Does your results nicely follow from adapting the REP-UCB algorithm Uehara et al, 2021 [2] with the CVaR bonus of Wang et al, 2023 [1]?

Also, if you consider time-homogenous transitions as was done in Uehara et al, 2021[2] do you match their dependency on $H$? Also do you think your dependence on $d$ and $H$ are tight? If yes, then why? If no, then where to do think this looseness arises from and how can it be improved?

[1] Wang, Kaiwen, Nathan Kallus, and Wen Sun. "Near-Minimax-Optimal Risk-Sensitive Reinforcement Learning with CVaR." arXiv preprint arXiv:2302.03201 (2023).

[2] Uehara, Masatoshi, Xuezhou Zhang, and Wen Sun. "Representation learning for online and offline rl in low-rank mdps." arXiv preprint arXiv:2110.04652 (2021).

---

> ### Author Response · Authors · 2023-11-17
> **Response to Reviewer i64s Part 1**
>
> We thank you for your valuable feedback and for appreciating our contributions. Below, we address your comments on our work.
>
> **Lower bounds and tightness**:  Establishing lower bounds for CVaR in low-rank MDPs is inherently challenging due to the complexity of the risk landscape and the non-linearity in function approximation methods used. Such lower bounds are context-dependent and often require additional assumptions that may not generalize well across different problem settings. Regarding standard episodic RL, which is a well-studied area, it's only recently that we have seen some results [3,4] on sample lower bounds in the context of low-rank MDPs.
>
> These results do not directly apply to the CVaR risk landscape; we believe it can be an important future direction. We guess the lower bound may take the form of $\Omega\left(\frac{HAd}{\tau \epsilon^2}\right)$ according to the lower bounds in low-rank MDPs and tabular CVaR RL [1]. If this guess is accurate, our algorithm has a loose dependency on $H$ and $\tau$ in the sample complexity. This is probably because we use the Hoeffding-type bonus instead of the Bernstein-type bonus. We will clarify this in the revised version.
>
> **Weird policy**:  We are sorry we are slightly confused about the review. $S_h$ is reached by executing action $a_1,\cdots,a_{h-1}$ following the policy $\pi_k$. After reaching $s_h$, we don’t follow $\pi_k$ and take two uniform actions $a_{h}, a_{h+1}$, i.e., $a_h \sim U(A), a_{h+1}\sim U(A)$. Therefore, $a_{h-1}$ is not a uniform action, and $s_h$ does not result from taking $a_{h-1}$ in $s_{h}$.
>
> **novelty**: We value the reviewer’s inquiry into the novelty of our work and acknowledge the relevance of the foundational studies by [1], [2]. Our work indeed builds upon these pivotal contributions but also introduces novel elements that we believe advance the field significantly.
> * Unlike [1], which studied tabular MDPs, in low-rank MDP, since the underlying representations are unknown, the problem becomes significantly harder because we need to balance representation learning, exploration, and exploration carefully. In Wang et al., 2023, there is no need to perform representation learning. Meanwhile, there is no need for efficient exploration because they can build model estimates in a state- and action-wise manner. Thus, their method cannot apply to function approximation with enormous state space. By dealing with the challenges, we answer the open problem raised by [1] affirmatively.
> * Work [2] studied discounted infinite-horizon MDP, where the model transitions are time-homogeneous, so they did not need to perform model estimation for every timestep. Besides, static risk measure is defined with finite steps, so their framework does not fit. This work studies the standard episodic MDP with static CVaR metric. We have two major technical difficulties.
>    * First, our subtle sampling process is built to provide time-dependent model transitions. Specifically, we need to carefully maintain two time-dependent datasets, allowing for a diversified exploration. Please see more details in our Response to Reviewer htXD.
>    * Second, strategic planning (i.e., updating policies) in [2] (cf. Line 9, Algorithm 1 therein) is achieved straightforwardly by vanilla value iteration. However, in stark contrast to standard risk-neutral RL, performing LSVI for the CVaR metric is computationally infeasible because of its non-linearity (cf. bi-level optimization structure in Equation 4). A significant remedy to this is our discretized planning oracle called CVaR-LSVI. Equipped with the oracle, our ELLA algorithm only requires a polynomial running time in addition to polynomial calls to the MLE oracle. As far as we are concerned, a computation- and sample-efficient planning oracle is missing in the current risk-averse RL community and can be widely adopted by following works.
>
> In light of the above points, we argue that our contributions are substantive and provide considerable value to the existing body of research. We will articulate these points more emphatically in the revised manuscript to better highlight the novel aspects of our research.
>
> [1] Wang, Kaiwen, Nathan Kallus, and Wen Sun. "Near-Minimax-Optimal Risk-Sensitive Reinforcement Learning with CVaR." arXiv preprint arXiv:2302.03201.
>
> [2] Uehara, Masatoshi, Xuezhou Zhang, and Wen Sun. "Representation learning for online and offline rl in low-rank mdps." arXiv preprint arXiv:2110.04652.
>
> [3] Cheng, Y., Huang, R., Yang, J., & Liang, Y. Improved Sample Complexity for Reward-free Reinforcement Learning under Low-rank MDPs. arXiv preprint arXiv:2303.10859.
>
> [4] Zhao, Canzhe, Ruofeng Yang, Baoxiang Wang, Xuezhou Zhang, and Shuai Li. "Learning Adversarial Low-rank Markov Decision Processes with Unknown Transition and Full-information Feedback." In Thirty-seventh Conference on Neural Information Processing Systems. 2023.

---

> ### Author Response · Authors · 2023-11-17
> **Response to Reviewer i64s Part 2**
>
> **if using time-homogenous transitions?** Thanks for raising this. Below, we provide clarifications on your inquiry.
>
> * Firstly, since we focus on CVaR RL, it is usually standard to study the classical finite-horizon episodic MDP setting, which admits that the transition structure is time-dependent. Time-homogenous transitions often appear in infinite-horizon discounted MDP. Therefore, they might not fit the CVaR objective of our interest.
> * Secondly, regarding our dependence on $d$ and $H$. Compared to [2], the dependence on the dimension $d$ is tight. For horizon $H$, we note that [2] assumes that the trajectory reward is normalized to 1 (this means $\sum_{h=1}^H r_h\le 1$ in the episodic setting), while we assume the immediate reward is bounded by $r_h \in [0,1]$. If we adopt this reward normalization, our PAC bound can be improved to $\widetilde{O}(d^4H^5)$, matching the rates.
> * Finally, if we need to adopt time-homogeneous transitions, our bound can be improved to $\widetilde{O}(d^4H^4)$.

---

### Official Review · Reviewer_htXD · 2023-11-04

**Soundness:** 3 good
**Presentation:** 3 good
**Contribution:** 3 good
**Rating:** 6
**Confidence:** 2

**Summary:**

This paper studies risk-sensitive RL in in low-rank MDPs with nonlinear function approximation, where the goals is to maximize the Conditional Value at Risk (CVaR). The authors proposed ELA (REprensentation Learning for CVAR) as shown in Algorithm 1, which contains a MLE oracle and uses UCB-based bonus to do exploration. They proved that the sample complexity of ELA is $\tilde{O}(1/\epsilon^2)$ The authors then proposed ELLA (REprensentation Learning with LSVI for CVAR) algorithm, which leverages least-squares value iteration to improve computationally efficiency.

**Strengths:**

1. The problem of study is interesting, i.e., risk-sensitive RL in in low-rank MDPs.
2. The presentation is clear and easy to understand.
3. Results seem reasonable and novel to me.

**Weaknesses:**

1. The proposed and improved algorithms are computationally inefficient in the sense that they still require to call MLE oracles.

**Questions:**

1. In Algorithm 1, for the part of collecting two transition tuples, could you explain why using different policies to sample the two actions.

---

> ### Author Response · Authors · 2023-11-17
> **Response to Reviewer htXD**
>
> We thank you for your valuable feedback and for appreciating our contributions. Below, we answer your questions about our work.
>
> **Data collection employed in Algorithm 1**: Thanks for raising this. The use of different policies to sample actions at this stage is intentional, allowing for a more diversified exploration of the model. By doing so, we maintain a balance between exploration and exploitation, a critical aspect in the context of risk-sensitive reinforcement learning. The two distinct datasets, $D_h$ and $D_h'$, have different (marginalized) distributions, serving separate purposes that contribute to our algorithm's proficiency. Specifically,
>  *  The first set, $D_h$, documents the progression of the exploration policy $\pi_{k−1}$ up to step $h$ when rolled out with the initial budget $c_{k−1}$. This procedure helps in designing the bonus terms, which are instrumental in calculating the exploration's effectiveness (Please refer to Lemma C.2 for bonus design rationale).
>  * The second set, $D_h'$, is aimed at enhancing the quality of our transition kernel estimates, which is fundamental for improving the overall algorithm's efficiency. By taking two consecutive uniform random actions, we introduce variability that leads to a distinct set of transitions (as further explained in Lemma E.3).
>
> **Call to MLE**: We thank the reviewer for highlighting the computational aspects of our algorithms involving MLE oracles. In response to this concern, we offer the following clarifications and future directions for refinement:
> * In theory, MLE oracles are widely used in model-based analyses (e.g., FLAMBE [4], CPPO [5], REP-UCB [6])
> * MLE oracles are also widely used in practice. For example, in RLHF, the reward model is trained based on a maximum likelihood estimator (MLE) [1,2]. They are also widely used in general RL [3]. In practice, the MLE oracle can be reasonably implemented whenever optimizing over $\mathcal{F}$ is feasible (e.g., parameterized by neural networks). Specifically, one can simply use gradient descent to minimize the empirical loss. Therefore, we think MLE oracles are feasible in practice.
>
> [1] Ouyang, L., Wu, J., Jiang, X., Almeida, D., Wainwright, C., Mishkin, P., ... & Lowe, R. (2022). Training language models to follow instructions with human feedback. Advances in Neural Information Processing Systems, 35, 27730-27744.
>
> [2] Christiano, P. F., Leike, J., Brown, T., Martic, M., Legg, S., & Amodei, D. (2017). Deep reinforcement learning from human preferences. Advances in neural information processing systems, 30.
>
> [3] Farahmand, A. M., Barreto, A., & Nikovski, D. (2017, April). Value-aware loss function for model-based reinforcement learning. In Artificial Intelligence and Statistics (pp. 1486-1494). PMLR.
>
> [4] Agarwal, A., Kakade, S., Krishnamurthy, A., & Sun, W. (2020). Flambe: Structural complexity and representation learning of low-rank mdps. Advances in neural information processing systems, 33, 20095-20107.
>
> [5] Uehara, M., & Sun, W. Pessimistic Model-based Offline Reinforcement Learning under Partial Coverage. ICLR 2022
>
> [6] Uehara, Masatoshi, Xuezhou Zhang, and Wen Sun. "Representation learning for online and offline rl in low-rank mdps." arXiv preprint arXiv:2110.04652 (2021).

---

### Author Response · Authors · 2023-11-21
**Summary of techniques**

We thank all the reviewers for reading the paper and providing valuable comments. Below, we summarize the techniques we used in the theoretical analysis.

*  **technical hardness of CVaR RL in low-rank MDPs**: We note that the primary challenge in addressing efficient exploration in CVaR-RL is managing the risk budget's *continuous* and *history-dependent* nature, which significantly deviates from the risk-neutral setting considered in the previous work and calls for novel techniques. In our response to Reviewer hVSJ, we have demonstrated the continuous nature of the risk budget in CVaR RL and our corresponding solution (discretized LSVI oracle). Below, we elaborate on the technical challenge that history-dependent nature brings.

     * The augmented policy and budget are history-dependent in the original MDPs, making it a complex task to bound the expectation of the estimation error of the transition kernel at timestep $h+1$ under the augmented policy, based only on statistics from timestep $h$ (Lemmas E.3 and E.4). This challenge is distinct from the more straightforward scenario in a risk-neutral setting, where the policy is Markovian and such a bounding process is more apparent.
      * To bypass the difficulty, we introduce the quantity $\omega_{h, \mathcal{P}}^{(\pi, c_1)}$ to keep track of the remaining budget, conditioned on which action $a_{h+1}$ is independent of the history, and build our new proof based on this new quantity.  This novel and important modification helps us overcome the history-dependent challenge. We believe the introduced risk budget distribution can be widely used in the following works.
      * Please find more details about the hardness in our response to Reviewer hVSJ.


* **analysis of the discretized planning oracle**: Secondly, our discretized planning oracle, CVaR-LSVI, requires new techniques.
   * First, we aim to show that LSVI works for CVaR with continuous reward. This is not trivial because the augmented MDP involves the evolution of the budget, which can be non-linear.
   * We need to establish a new concentration inequality for CVaR-LSVI (Lemma D.1) because we introduce the new budget variable in the regression and need to compute its covering number. We believe this oracle's design can benefit the risk-averse RL community.

To summarize, we assert that our techniques provide meaningful advancements and significantly enhance the current scope of research in this field. In the next version, we will highlight these technical contributions. We are more than happy to clarify any concerns.

---

### Meta-Review · Area_Chair_dwxo · 2023-12-07

**Metareview:**

The authors derive regret bounds for low-rank MDPs where the representation is unknown.

The reviewers find the problem and setting novel and interesting.
The paper is well executed, and it is important to scale up CVaR to more complex function classes (such as the one introduced here).
Moreover, the algorithm is oracle efficent.
However, there are some algorithmic similarities with the work of Uheara cited by the reviewers.
The tightness of the bound is not investigated, however (e.g., no lower bound is presented)

**Justification For Why Not Higher Score:**

The paper examines a setting of somewhat limited audience.

**Justification For Why Not Lower Score:**

The paper is well executed  and examines a setting which is interesting and previously unexplored, and bring to the community valuable results

---

### Decision · Program_Chairs · 2024-01-16

Accept (poster)